# WHICH MODE IS BETTER FOR FEDERATED LEARNING? CENTRALIZED OR DECENTRALIZED

## ABSTRACT

Both centralized and decentralized approaches have shown excellent performance and great application value in federated learning (FL). However, current studies do not provide sufficient evidence to show which one performs better. Although from the optimization perspective, decentralized methods can approach the comparable convergence of centralized methods with less communication, its test performance has always been inefficient in empirical studies. To comprehensively explore their behaviors in FL, we study their excess risks, including the joint analysis of both optimization and generalization. We prove that on smooth non-convex objectives, 1) centralized FL (CFL) always generalizes better than decentralized FL (DFL); 2) from perspectives of the excess risk and test error in CFL, adopting partial participation is superior to full participation; and, 3) there is a necessary requirement for the topology in DFL to avoid performance collapse as the training scale increases. Based on some simple hardware metrics, we could evaluate which framework is better in practice. Extensive experiments are conducted on common setups in FL to validate that our theoretical analysis is contextually valid in practical scenarios.

## 1 INTRODUCTION

Since McMahan et al. (2017) propose FL, it becomes a promising paradigm for training the heterogeneous dataset. Classical `FedAvg` utilizes the CFL framework consisting of a global server and massive local clients, to jointly train a global model via periodic communication. Though it shines in large-scale training, CFL has to afford expensive communication costs under a large number of local clients. To efficiently alleviate this pressure, DFL is introduced as a compromise. Sun et al. (2022) propose the `DFedAvg` and analyze its fundamental properties, which adopts one specific topology across all clients to significantly reduce the number of links for communication. In some scenarios, it is impossible to set up the global server, in which case DFL becomes the only valid solution. As two major frameworks in the current FL community, both centralized and decentralized approaches are well-studied and improved greatly. More and more insights are being revealed to give it huge potential for applications in practice. However, there is still a *question* lingering in the studies:

> *Which framework is better for federated learning? Centralized or decentralized?*

Research on this question goes back to distributed learning based on the parallel stochastic gradient descent (PSGD). Lian et al. (2017) study the comparison between centralized PSGD (C-PSGD) and decentralized PSGD (D-PSGD), and provide a positive answer for decentralized approaches. D-PSGD can achieve a comparable convergence rate with linear speedup as the C-PSGD with much fewer communication links. However, in the FL framework, each local dataset follows the unknown heterogeneous distribution, which leads to consistently poor experimental performance in DFL. Shi et al. (2023) also explore that an algorithm often performs worse on the DFL framework under the same experimental setups. Although some studies provide consequential illustrations based on the analysis of spectral gaps and consensus, we still don't know the clear answer to the *question* above.

Most of the previous works focus on the analysis of convergence rates and ignore the generalization efficiency, while the test accuracy is highly related to the generalization. Therefore, the incomplete comparison can easily lead to cognitive misunderstandings. In order to further understand their performance and comprehensively answer the above *question*, we follow Zhou et al. (2021); Sun et al. (2023d) to introduce the excess risk as the measurements of test errors, which analyzes the joint impacts of both optimization and generalization. Meanwhile, we utilize the uniform stability (Elisseeff et al., 2005; Hardt et al., 2016) to learn their generalization abilities. To improve the applicability of

Table 1: Main results on the excess risk of `FedAvg` and `DFedAvg` on the smooth non-convex objective. $\kappa_\lambda$ is a constant related to spectrum gap $1 - \lambda$ (Spectrum norm $\lambda$ is defined in Lemma 1).

| | Excess Risk $\varepsilon = \varepsilon_O + \varepsilon_G$ | Optimal Selection | Corresponding $\varepsilon_G$ |
|---|---|---|---|
| `FedAvg` | $\mathcal{O}\left(\frac{1}{nKT} + \frac{1}{S}\frac{n^{\frac{\mu L}{1+\mu L}}}{m}(KT)^{\frac{\mu L}{1+\mu L}}\right)$ | $n = \mathcal{O}\left(m^{\frac{1+\mu L}{1+2\mu L}}\right)$ | $\mathcal{O}\left(m^{-\frac{1+\mu L}{1+2\mu L}}\right)$ |
| `DFedAvg` | $\mathcal{O}\left(\frac{1}{T} + \frac{1}{S}\left(\frac{1+6\sqrt{m}\kappa_\lambda}{m}\right)^{\frac{1}{1+\mu L}}(KT)^{\frac{\mu L}{1+\mu L}}\right)$ | $\kappa_\lambda = 0$ | $\mathcal{O}\left(m^{-\frac{1}{1+\mu L}}\right)$ |

$n$: the number of clients participating in the training per round in CFL; $m$: the number of total clients; $K$: local interval; $T$: communication rounds; $L$: Lipschitz constant; $\mu$: specific constant with $\mu \leq 1/L$.

the vanilla uniform stability analysis on general deep models, we remove the idealized assumption of bounded full gradients and adopt the bounded uniform stability instead.

Our work provides a novel and comprehensive understanding of comparisons between CFL and DFL as shown in Table 1. We analyze excess risks for the classical `FedAvg` and `DFedAvg`. Both of them achieve the same results as vanilla SGD (Hardt et al., 2016) on the impact of iterations $TK$. Vanilla SGD could achieve $\mathcal{O}(\frac{1}{Sm})$ on total $Sm$ data samples. However, due to the local training process, both CFL and DFL are slower than this rate in generalization. The generalization achieves $\mathcal{O}(\frac{1}{S}\frac{n^{\frac{\mu L}{1+\mu L}}}{m})$ in CFL and $\mathcal{O}(\frac{1}{S}\left(\frac{1+6\sqrt{m}\kappa_\lambda}{m}\right)^{\frac{1}{1+\mu L}})$ in DFL respectively. From the generalization perspective, CFL always generalizes better than DFL. From the excess risk and test error perspectives, to achieve optimal performance, CFL only needs partial clients to simultaneously participate in the training per communication round. Our analysis not only quantifies the differences between them but also demonstrates some discussions on the negative impact of topology in DFL. As a compromise of CFL to save communication costs, the topology adopted in DFL has a specific minimum requirement to avoid performance collapse when the number of local clients increases. We also conduct extensive experiments on the widely used FL setups to validate our analysis. Both theoretical and empirical studies confirm the validity of our answers to the *question* above.

We summarize the main contributions of this paper as follows:

- We provide the uniform stability and excess risks analysis for the `FedAvg` and `DFedAvg` algorithms without adopting the idealized assumption of bounded full gradients.

- We prove centralized approaches always generalize better than decentralized ones, and CFL only needs partial participation to achieve optimal test error. Furthermore, we can estimate at what scale of devices, CFL would be suitable under the acceptable communication costs.

- We prove even with adopting DFL as a compromise of CFL, there is a minimum requirement on the topology. Otherwise, even with more local clients and training data samples, its generalization performance still gets worse.

- We conduct empirical studies to validate our theoretical analysis and support our answer.

## 2 RELATED WORK

**Centralized federated learning.** Since McMahan et al. (2017) propose the fundamental CFL algorithm `FedAvg`, several studies explore its strengths and weaknesses. Yang et al. (2021) prove its convergence rate on the non-convex and smooth objectives satisfies the linear speedup property. Furthermore, Karimireddy et al. (2020) study the client-drift problems in FL and adopt the variance reduction technique to alleviate the local overfitting. Li et al. (2020) introduce the proxy term to force the local models to be close to the global model. Zhang et al. (2021); Acar et al. (2021); Gong et al. (2022); Sun et al. (2023b) study the primal-dual methods in CFL and prove it achieves faster convergence. This is also the optimal convergence rate that can be achieved in the current algorithms. With the deepening of research, researchers begin to pay attention to its generalization ability. One of the most common analyses is the PAC-Bayesian bound. Yuan et al. (2021) learn the components in the generalization and formulate them as two expectations. Reisizadeh et al. (2020) define the margin-based generalization error of the PAC-Bayesian bound. Qu et al. (2022); Caldarola et al. (2022); Sun et al. (2023b) study the generalization efficiency in CFL via local sharpness aware minimization. Sun et al. (2023a) re-define the global margin-based generalization error and discuss the differences between local and global margins. Sefidgaran et al. (2023) also learn the

reduction of the communication may improve the generalization performance. In addition, uniform stability (Elisseeff et al., 2005; Hardt et al., 2016) is another powerful tool adopted to measure the generality. Yagli et al. (2020) learn the generalization error and privacy leakage in federated learning via the information-theoretic bounds. Sun et al. (2023e) provide the stability analysis for several FL methods. Sun et al. (2023d) prove stability in CFL is mainly affected by the consistency.

**Decentralized approaches.** Since Wang & Joshi (2021) learn a unified framework on the local-updates-based methods, Yuan et al. (2020) explore the impact of the bias correction in a distributed framework and reveal its efficiency in the training. Based on (Shi et al., 2015) which analyzes the consensus in decentralized approaches, Alghunaim & Yuan (2022) provide a novel unified analysis for the non-convex decentralized learning. After that, local updates also draw much attention to efficient training. Mishchenko et al. (2022); Nguyen et al. (2022); Alghunaim (2023) learn the advantages in the local process in both centralized and decentralized approaches.

**Decentralized federated learning.** Since Lalitha et al. (2018) propose the prototype of DFL, it is becoming a promising approach as the compromise of CFL to save the communication costs. Lian et al. (2018); Yu et al. (2019); Assran et al. (2019); Koloskova et al. (2020) learn the stability of decentralized SGD which contributes to the research on the heterogeneous dataset. Hu et al. (2019) study the gossip communication and validate its validity. Hegedűs et al. (2021) explore the empirical comparison between the prototype of DFL and CFL. Lim et al. (2021) propose a dynamic resource allocation for efficient hierarchical federated learning. Sun et al. (2022) propose the algorithm `DFedAvg` and prove that it achieves the comparable convergence rate as the vanilla SGD method. Gholami et al. (2022) also learn the trusted DFL framework on the limited communications. Hashemi et al. (2021); Shi et al. (2023) verify that DFL suffers from the consensus and may be improved by multi-gossip. Li et al. (2023) propose the adaptation of the variant of the primal-dual optimizer in the DFL framework. However, experiments in DFL are generally unsatisfactory. Research on its generalization has gradually become one of the hot topics. Sun et al. (2021) provide the uniform stability analysis of the decentralized approach and indicate that it could be dominated by the spectrum coefficient. Zhu et al. (2023) prove the decentralized approach may be asymptotically equivalent to the SAM optimizer with flat loss landscape and higher generality. Different from the previous work, our study mainly focuses on providing a clear and specific answer to the *question* we ask in Section 1. Meanwhile, our analysis helps to understand whether a topology is suitable for DFL, and how to choose the most suitable training mode under existing conditions in practice.

## 3 PROBLEM FORMULATION

**Notations.** We first introduce some notations and marks adopted in our paper as follows and in Table 2. In this paper, unless otherwise specified, we use italics for scalars, e.g. $n$, and capital boldface for matrix, e.g. $\mathbf{M}$. $[n]$ denotes a sequence of positive integers from 1 to $n$. $\mathbb{E}[\cdot]$ denotes the expectation of $\cdot$ term with respect to the potential

Table 2: Notation tables.

| Symbol | Definition |
|--------|------------|
| $m$ | number of total clients in CFL/DFL |
| $n$ | number of participating clients in CFL |
| $\mathcal{G}$ | topology connection in DFL |
| $U$ | upper bound of loss value |

probability spaces. $\|\cdot\|$ denotes the $l_2$-norm of a vector and the Frobenius norm of a matrix. $\|\cdot\|_{op}$ denotes the spectral norm of a matrix. $|\cdot|$ denotes the absolute value of a scalar. Unless otherwise specified, all four arithmetic operators conform to element-wise operations.

**Fundamental Problems.** We formulate the fundamental problem as minimizing a finite-sum problem with privacy on each local heterogeneous dataset. We suppose the following simplified scenario: a total of $m$ clients jointly participate in the training whose indexes are recorded as $i$ where $i \in [m]$. On each client $i$, there is a private dataset $\mathcal{S}_i$ with $S$ data samples. Each sample is denoted as $z_{i,j}$ where $j \in [S]$. Each dataset $\mathcal{S}_i$ follows a different and independent distribution $\mathcal{D}_i$. We consider the population risk minimization on the finite-sum problem of several non-convex objectives $f_i(w, z)$:

$$\min_{w} F(w) \triangleq \frac{1}{m} \sum_{i \in [m]} F_i(w), \quad F_i(w) \triangleq \mathbb{E}_{z \sim \mathcal{D}_i} f_i(w, z), \tag{1}$$

where $F(w) : \mathbb{R}^d \to \mathbb{R}$ is denoted as the global objective with respect to the parameters $w$. In the general cases, we use the surrogate empirical risk minimization (ERM) objective to replace Eq.(1):

$$\min_{w} f(w) \triangleq \frac{1}{m} \sum_{i \in [m]} f_i(w), \quad f_i(w) \triangleq \frac{1}{S} \sum_{z \in \mathcal{S}_i} f_i(w, z). \tag{2}$$

**Centralized FL.** Centralized federated learning employs a global server to coordinate several local clients to collaboratively train a global model. To alleviate the communication costs, it randomly activates a subset $\mathcal{N}$ $(|\mathcal{N}| = n)$ among all clients. At the beginning of each round, the global server sends the global model to the active clients as the initialization state. Then they will train the model on their local dataset. After the local training, the optimized local models will be sent to the global server for aggregation. The aggregated model will become the global model in the next round and continue to participate in training until it is well optimized. Algorithm 1 shows the classical `FedAvg` method (McMahan et al., 2017).

---

**Algorithm 1:** `FedAvg` Algorithm

**Input:** initial model $w^0, T, K, \eta$
**Output:** optimized global model $w^T$.

1 **for** $t = 0, 1, ..., T-1$ **do**
2      randomly select a subset $\mathcal{N}$ from $[m]$
3      **for** $i \in \mathcal{N}$ **in parallel do**
4          send $w^t$ to the client $i$ as $w_{i,0}^t$
5          $w_{i,K}^t \leftarrow$ SGD-Opt$(w_{i,0}^t, \eta, K)$
6          send the $w_{i,K}^t$ to the server
7      **end**
8      $w^{t+1} \leftarrow \frac{1}{n} \sum_{i \in \mathcal{N}} w_{i,K}^t$
9 **end**

---

**Decentralized FL.** Decentralized FL allows each local client to only communicate with its neighbors on an undirected graph $\mathcal{G}$, which is defined as a collection of clients and connections between clients $\mathcal{G} = (\mathcal{I}, \mathcal{E})$. $\mathcal{I}$ denotes the clients' set $[m]$ and $\mathcal{E} \subseteq \mathcal{I} \times \mathcal{I}$ denotes the their connections. The relationships are associated with an adjacent matrix $\mathbf{A} = [a_{ij}] \in \mathbb{R}^{m \times m}$. If $(i, j) \in \mathcal{E}$, the corresponding element $a_{ij} > 0$, otherwise $a_{ij} = 0$. In the decentralized setting, all clients first aggregate the models within their neighborhoods and then train them on their local dataset. Algorithm 2 shows the classical `DFedAvg` (Sun et al., 2022).

---

**Algorithm 2:** `DFedAvg` Algorithm

**Input:** initial models $w_{i,K}^{-1}, T, K, \eta, \mathcal{G}$
**Output:** optimized global model $w^T$.

1 **for** $t = 0, 1, ..., T-1$ **do**
2      **for** $i \in [m]$ **in parallel do**
3          send $w_{i,K}^{t-1}$ to its neighbors
4          $w_{i,0}^t \leftarrow \sum_{j \in \mathcal{A}_i} a_{ij} w_{i,K}^{t-1}$
5          $w_{i,K}^t \leftarrow$ SGD-Opt$(w_{i,0}^t, \eta, K)$
6      **end**
7 **end**

---

**Definition 1 (Adjacent Matrix)** *The adjacent matrix $\mathbf{A} = [a_{ij}] \in \mathbb{R}^{m \times m}$ satisfies the following properties: (1) non-negative: $a_{ij} \geq 0$; (2) symmetry: $\mathbf{A}^\top = \mathbf{A}$; (3) null $\{\mathbf{I} - \mathbf{A}\} = span\{1\}$; (4) spectral: $\mathbf{I} \succeq \mathbf{A} \succ -\mathbf{I}$; (5) double stochastic: $\mathbf{1}^\top \mathbf{A} = \mathbf{1}^\top, \mathbf{A} \cdot \mathbf{1} = \mathbf{1}$ where $\mathbf{1} = [1, 1, 1, \cdots, 1]^\top$.*

**Lemma 1** *The eigenvalues $\lambda_i$ of matrix $\mathbf{A}$ satisfies $1 = \lambda_1 > \lambda_2 > \cdots > \lambda_m > -1$, where $\lambda_i$ denotes the $i$-th largest eigenvalue of $\mathbf{A}$. By defining $\lambda \triangleq \max\{|\lambda_2|, |\lambda_m|\} > 0$, we can bound the spectral gap of the adjacent matrix $\mathbf{A}$ by $\lambda$, which could measure the connections of this topology.*

**Lemma 2 ((Montenegro et al., 2006))** *Let the matrix $\mathbf{P} = \mathbf{1}\mathbf{1}^\top/m \in \mathbb{R}^{m \times m}$, given a positive $t \in \mathbb{Z}^+$, the adjacent matrix $\mathbf{A}$ satisfies $\|\mathbf{A}^t - \mathbf{P}\|_{op} \leq \lambda^t$, which measures the ability of aggregation.*

**Generalization Gap.** Because of the unknown distribution $\mathcal{D}_i$ in Eq.(1), we often generate the optimized solution with Eq.(2). Therefore, there is an inevitable gap between the expected solution and the obtained one. An intuitive example is that the model finely trained on the training set often performs poorly on the test set, which leads to overfitting. This also makes studying how generalization gaps are affected by the training process a key challenge in the field of machine learning. We denote $\mathcal{C}$ as the joint dataset of total local dataset $\mathcal{S}_i$. We consider using the specific random algorithm $\mathcal{A}$ to solve the Eq.(2) on the joint dataset $\mathcal{C}$ and obtain the solution $\mathcal{A}(\mathcal{C})$. Therefore, the gap could be defined as the generalization error $\varepsilon_G = \mathbb{E}_{\mathcal{C}, \mathcal{A}}[F(\mathcal{A}(\mathcal{C})) - f(\mathcal{A}(\mathcal{C}))]$. It reflects the joint impact caused by the dataset $\mathcal{C}$ and the algorithm $\mathcal{A}$. As an important term of the excess risk error, the generalization error $\varepsilon_G$ demonstrates the potential instability of the proposed algorithm. To comprehensively explore the impacts, motivated by the previous studies (Hardt et al., 2016; Sun et al., 2021; Zhou et al., 2021; Sun et al., 2023d;e), we adopt the uniform stability analysis.

**Definition 2 (Uniform Stability)** *We construct a new joint dataset $\widetilde{\mathcal{C}}$ which only differs from the vanilla dataset $\mathcal{C}$ at most one data sample. Then we say $\mathcal{A}$ is an $\epsilon$-uniformly stable algorithm if:*

$$\sup_{z \sim \cup \mathcal{D}_i} \mathbb{E}\left[f(\mathcal{A}(\mathcal{C}), z) - f(\mathcal{A}(\widetilde{\mathcal{C}}), z)\right] \leq \epsilon. \tag{3}$$

**Lemma 3** *(Elisseeff et al., 2005; Hardt et al., 2016) If the stochastic algorithm $\mathcal{A}$ is $\epsilon$-uniformly stable, we could bound its generalization error as $\varepsilon_G \leq \epsilon$ on the corresponding dataset distribution.*

**Excess Risk.** According to the definition of $\varepsilon_G$, it represents the performance errors on the optimized model $\mathcal{A}(\mathcal{C})$ between the training samples and the real data distribution. However, the real test error also should include the error from the $\mathcal{A}(\mathcal{C})$ itself, which is, the error between the optimized model $\mathcal{A}(\mathcal{C})$ and the real optimum $w^\star$. Therefore, we introduce the excess risk to measure the error of the test accuracy. Generally, it could be approximated as the term $\mathbb{E}[F(\mathcal{A}(\mathcal{C}))]$ instead.

**Definition 3 (Excess Risk)** *We denote the $w^\star$ as the true optimum which can be achieved by the algorithm $\mathcal{A}$ on the dataset $\mathcal{C}$. $\mathbb{E}[f(w^\star)]$ is usually very small. Thus the excess risk is defined as:*

$$\varepsilon = \mathbb{E}\left[F(\mathcal{A}(\mathcal{C})) - f(w^\star)\right] = \underbrace{\mathbb{E}\left[F(\mathcal{A}(\mathcal{C})) - f(\mathcal{A}(\mathcal{C}))\right]}_{\varepsilon_G} + \underbrace{\mathbb{E}\left[f(\mathcal{A}(\mathcal{C})) - f(w^\star)\right]}_{\varepsilon_O}. \tag{4}$$

The excess risk can effectively measure the true test performance $\mathbb{E}[F(\mathcal{A}(\mathcal{C}))]$, which is also one of the very important concerns in current studies of machine learning. Many previous studies have studied the optimization error $\varepsilon_O$ of general centralized and decentralized federated learning frameworks. Our work mainly focuses on the generalization error bounds for these two frameworks and we also provide a comprehensive theoretical analysis of their excess risks in the Section 4.

## 4 THEORETICAL ANALYSIS

In this part, we mainly introduce the theoretical analysis of the generalization error bound and provide a comparison between centralized and decentralized setups in FL paradigms. We first introduce the main assumptions adopted in this paper and discuss their applicability and our improvements compared with previous studies. Then we state the main theorems, corollaries, and discussions.

### 4.1 ASSUMPTIONS

**Assumption 1** *For $\forall\, w_1, w_2 \in \mathbb{R}^d$, the objective $f_i(w)$ is L-smooth for arbitrary data sample z:*

$$\|\nabla f_i(w_1, z) - \nabla f_i(w_2, z)\| \leq L\|w_1 - w_2\|. \tag{5}$$

**Assumption 2** *For $\forall\, w \in \mathbb{R}^d$, the stochastic gradient $g_i = \nabla f_i(w, z)$ where $z \in \mathcal{S}_i$ is an unbiased estimator of the full local gradient $\nabla f_i(w) \triangleq \mathbb{E}_{z \sim \mathcal{D}_i}[\nabla f_i(w, z)]$ with a bounded variance, i.e.,*

$$\mathbb{E}_z[\, g_i - \nabla f_i(w)] = 0, \ \mathbb{E}_z\|g_i - \nabla f_i(w)\|^2 \leq \sigma_l^2. \tag{6}$$

**Assumption 3** *For $\mathcal{A}(\mathcal{C}), \mathcal{A}(\widetilde{\mathcal{C}}) \in \mathbb{R}^d$ which are well trained by an $\epsilon$-uniformly stable algorithm $\mathcal{A}$ on dataset $\mathcal{C}$ and $\widetilde{\mathcal{C}}$, the global objective $f(w)$ satisfies G-Lipschitz continuity between them, i.e.,*

$$|f(\mathcal{A}(\mathcal{C})) - f(\mathcal{A}(\widetilde{\mathcal{C}}))| \leq G\|\mathcal{A}(\mathcal{C}) - \mathcal{A}(\widetilde{\mathcal{C}})\|. \tag{7}$$

**Discussions.** Assumption 1 and 2 are two general assumptions that are widely adopted in the analysis of federated stochastic optimization (Reddi et al., 2020; Karimireddy et al., 2020; Gorbunov et al., 2021; Yang et al., 2021; Xu et al., 2021; Gong et al., 2022; Qu et al., 2022; Sun et al., 2023c; Huang et al., 2023). Assumption 3 is a variant of the vanilla Lipschitz continuity assumption. The vanilla Lipschitz continuity is widely used in the uniform stability analysis (Elisseeff et al., 2005; Hardt et al., 2016; Zhou et al., 2021; Sun et al., 2021; Xiao et al., 2022; Zhu et al., 2022; Sun et al., 2023d). Vanilla Lipschitz continuity assumption implies the objective have bounded gradients $\|\nabla f(w)\| \leq G$ for $\forall\, w \in \mathbb{R}^d$. However, several recent works have shown that it may not always hold in current deep learning (Kim et al., 2021; Mai & Johansson, 2021; Patel & Berahas, 2022; Das et al., 2023). Therefore, in order to improve the applicability of the stability analysis on general deep models, we use Assumption 3 instead, which could be approximated as a specific Lipschitz continuity only at the minimum $\mathcal{A}(\mathcal{C})$. The main challenge without the strong assumption is that the boundedness of the iterative process cannot be ensured. Our proof indicates that even if the iterative process is not necessarily bounded, the uniform stability can still maintain the vanilla upper bound.

### 4.2 STABILITY OF CENTRALIZED FEDERATED LEARNING

In this part, we introduce the stability of the centralized federated learning setup. We consider the classical `FedAvg` (Algorithm 1) and analyze its stability during the training process.

**Theorem 1** *Under Assumption 1∼ 3, let the active ratio per communication round be $n/m$, and let the learning rate $\eta = \mathcal{O}\left(\frac{1}{tK+k}\right) = \frac{\mu}{tK+k}$ is decayed per iteration $\tau = tK + k$ where $\mu$ is a specific*

*constant which satisfies $\mu \leq \frac{1}{L}$, let U be the maximization of loss value, the stability of centralized federated learning satisfies:*

$$\mathbb{E}|f(w^{T+1};z) - f(\widetilde{w}^{T+1};z)| \leq \frac{2\sigma_l G}{mSL}\left(\frac{TK}{\tau_0}\right)^{\mu L} + \frac{nU\tau_0}{mS}. \tag{8}$$

*By selecting a proper $\tau_0 = \left(\frac{2\sigma_l G}{nUL}\right)^{\frac{1}{1+\mu L}}(TK)^{\frac{\mu L}{1+\mu L}}$, we can minimize the error bound:*

$$\mathbb{E}|f(w^{T+1};z) - f(\widetilde{w}^{T+1};z)| \leq \frac{4}{S}\left(\frac{\sigma_l G}{L}\right)^{\frac{1}{1+\mu L}}\left(\frac{n^{\frac{\mu L}{1+\mu L}}}{m}\right)(UTK)^{\frac{\mu L}{1+\mu L}}. \tag{9}$$

**Corollary 1.1 (Stability.)** *The stability of the centralized federated learning (CFL) is mainly affected by the number of samples $S$, the number of total clients $m$, the number of active clients $n$, and the total iterations $TK$. The vanilla SGD achieves the upper bound of $\mathcal{O}\left(T^{\frac{\mu L}{1+\mu L}}/S\right)$ (Hardt et al., 2016), while stability of CFL is worse than $\mathcal{O}\left(T^{\frac{\mu L}{1+\mu L}}/S\right)$ due to additional negative impacts of $n$ and $K$. Specifically, when the number of active clients and local intervals increases, its performance will decrease significantly. An intuitive understanding is that when the number of active clients increases, it will be easier to select new samples that are not consistent with the current understanding. This results in the model having to make updates to adapt to the new knowledge.*

**Corollary 1.2 (Excess Risk.)** *Haddadpour & Mahdavi (2019); Zhou et al. (2021) provide the analysis of $\varepsilon_O = \mathbb{E}\left[f(w^T) - f(w^\star)\right]$ under PŁ-condition. The convergence rate of $\texttt{FedAvg}$ is dominated by $\mathcal{O}(1/nKT)$ rate on non-convex smooth objectives. Therefore, when the number of dataset samples $S$ is fixed, the excess risk of CFL is dominated by $\mathcal{O}\left(1/nKT + (nKT)^{\frac{\mu L}{1+\mu L}}/m\right)$. Both terms are caused by the stochastic variance $\sigma_l$. Discussions are stated as follows.*

When $m = n = K = 1$, it degenerates into the conclusion of the vanilla SGD. When $m = n$ and $K > 1$, it degenerates into the conclusion of the local SGD. The analysis in centralized federated learning requires the $m \geq n > 1$ and $K > 1$. In Corollary 1.2, our analysis points out it is a trade-off on $n$, $K$, and $T$ in $\texttt{FedAvg}$ and centralized federated learning. We meticulously provide the recommended selection for the number of active clients $n$ to achieve optimal efficiency.

Generally, the communication round $T$ is decided by the training costs and local computing power. Therefore, under a fixed local interval $K$, increasing partial participation rates (increasing $n$) may hurt the final test performance. When the optimization error $\varepsilon_O$ dominates the excess risk, increasing $n$ brings linear speedup property and effectively reduces the optimization error (Yang et al., 2021). Most of the advanced federated methods have also been proven to benefit from this speedup. However, when the generalization error dominates the excess risk, large $n$ is counterproductive. Charles et al. (2021) similarly observe the experiments that larger cohorts (larger $n$) uniformly lead to worse generalization. We theoretically prove this phenomenon and provide a rough estimation of the best value of active ratios. When the number of total clients $m$ is fixed, the optimal number of the active clients in centralized federated learning satisfies $n^\star = \mathcal{O}(m^{\frac{1+\mu L}{1+2\mu L}})$, which could efficiently balance the optimization and generalization error. When jointly considering the impact of the local interval, the optimal value could be adjusted as $n^\star = \mathcal{O}(m^{\frac{1+\mu L}{1+2\mu L}}/K)$. We summarize them as follows:

- The best number of active clients per round is approximately in the order of $(m^{\frac{2}{3}}, m)$.

- When local intervals increase, decreasing $n$ appropriately can achieve better performance.

Empirical studies on centralized federated learning are shown in Section 5.1 and Appendix A.2.1.

### 4.3 STABILITY OF DECENTRALIZED FEDERATED LEARNING

In this part, we introduce the stability of the decentralized federated learning setup. We consider the classical $\texttt{DFedAvg}$ (Algorithm 2) and analyze its excess risk in the whole training process.

**Theorem 2** *Under Assumption 1$\sim$ 3, let the communication graph be $\mathbf{A}$ as introduced in Definition 1 which satisfies the conditions of spectrum gap $\lambda$ in the Lemma 1 and 2, and let the learning rate $\eta = \mathcal{O}\left(\frac{1}{tK+k}\right) = \frac{\mu}{tK+k}$ is decayed per iteration $\tau = tK + k$ where $\mu$ is a specific constant which satisfies $\mu \leq \frac{1}{L}$, let U be the maximization of loss value, the stability of decentralized*

*federated learning (Algorithm 2) satisfies:*

$$\mathbb{E}\left[|f(w^{T+1};z) - f(\widetilde{w}^{T+1};z)|\right] \leq \frac{2\sigma_l G}{SL}\left(\frac{1 + 6\sqrt{m}\kappa_\lambda}{m}\right)\left(\frac{TK}{\tau_0}\right)^{\mu L} + \frac{U\tau_0}{S}. \quad (10)$$

*where $\kappa_\lambda \approx \mathcal{O}\left(\frac{1}{\lambda \ln \frac{1}{\lambda}}\right)$ is a constant coefficient related to the spectrum norm $\lambda$.*

*By selecting a proper $\tau_0 = \left(\frac{2\sigma_l G}{UL}\frac{1+6\sqrt{m}\kappa_\lambda}{m}\right)^{\frac{1}{1+\mu L}}(TK)^{\frac{\mu L}{1+\mu L}}$, we can minimize the error bound:*

$$\mathbb{E}|f(w^{T+1};z) - f(\widetilde{w}^{T+1};z)| \leq \frac{4}{S}\left(\frac{\sigma_l G}{L}\right)^{\frac{1}{1+\mu L}}\left(\frac{1 + 6\sqrt{m}\kappa_\lambda}{m}\right)^{\frac{1}{1+\mu L}}(UTK)^{\frac{\mu L}{1+\mu L}}. \quad (11)$$

**Corollary 2.1 (Stability.)** *The stability of decentralized federated learning (DFL) performs with the impact of the number of samples $S$, the number of total clients $m$, and total iterations $TK$. Differently, it is also affected by the topology. $\kappa_\lambda$ is also a widely used coefficient related to the $\lambda$ that could measure different connections in the topology. Its stability achieves the best performance when we select the $\kappa_\lambda = 0$, which corresponds to the fully connected topology. In practical scenarios, DFL prefers a small $\kappa_\lambda$ coefficient to improve generalization performance as much as possible.*

**Corollary 2.2 (Excess Risk.)** *Similarly, we need to comprehensively consider the performance of the excess risk in the DFL framework. Haddadpour & Mahdavi (2019); Sun et al. (2022) study the convergence under the PŁ-condition and provide the analysis that the spectrum gap $\lambda$ generally exists in the non-dominant term, which indicates the optimization convergence achieves $\mathcal{O}(1/T)$ under the sufficiently large communication round $T$. Therefore, the excess risk of DFL is dominated by $\mathcal{O}\left(1/T + ((1 + 6\sqrt{m}\kappa_\lambda)/m)^{\frac{1}{1+\mu L}}(KT)^{\frac{\mu L}{1+\mu L}}\right)$. Further discussions are stated as follows.*

Corollary 2.2 explains that the spectrum gap mainly affects $\varepsilon_G$. Similarly, we consider the communication rounds $T$ to be determined by local computing power. When the local interval $K$ is fixed, selecting the topology with larger $\kappa_\lambda$ will achieve a bad performance. As stated in Corollary 2.1, when $\kappa_\lambda = 0$ it achieves a minimal upper bound. This also indicates that the fully connected topology is the best selection. According to the research of Ying et al. (2021), we show some classical topologies in Table 3 to compare their generalization performance. The classical topologies, i.e. `grid`, `ring`, and `star`, will lead to a significant

Table 3: Comparison of common topologies. The arrow denotes the trends as $m$ increases. $\widetilde{\mathcal{O}}(m)$ means the order of $m^{\frac{1}{1+\mu L}}$.

| Topologies | $\kappa_\lambda$ | $\varepsilon_G$ |
|---|---|---|
| `full` | $0$ | $\widetilde{\mathcal{O}}\left(m^{-1}\right)\downarrow$ |
| `exp` | $\mathcal{O}\left(\ln m\right)$ | $\widetilde{\mathcal{O}}\left(m^{-0.5}\right)\downarrow$ |
| `grid` | $\mathcal{O}\left(m\ln m\right)$ | $\widetilde{\mathcal{O}}\left(m^{0.5}\right)\uparrow$ |
| `ring` | $\mathcal{O}\left(m^2\right)$ | $\widetilde{\mathcal{O}}\left(m^{1.5}\right)\uparrow$ |
| `star` | $\mathcal{O}\left(m^2\right)$ | $\widetilde{\mathcal{O}}\left(m^{1.5}\right)\uparrow$ |

drop in stability as total clients $m$ increase. In contrast, the `full` and `exp` topologies show better performance which can reduce the generalization error as $m$ increases. However, small $\kappa_\lambda$ always means more connections in the topology and more communication costs, which is similar to the spectrum gap $1 - \lambda$ being large enough ($\lambda$ is small enough). This also gives us a clear insight into the design of the topology. We summarize the conclusions as follows:

- A better topology must satisfy the spectrum coefficient $\kappa_\lambda$ is small enough.

- To avoid performance collapse as $m$ increases, the topology should satisfy $\kappa_\lambda \leq \mathcal{O}\left(\sqrt{m}\right)$.

Empirical studies on decentralized federated learning are shown in Section 5.2 and Appendix A.2.2.

### 4.4 COMPARISONS BETWEEN CFL & DFL

In this section, we mainly discuss the strengths and weaknesses of each framework. In Table 1, we summarize Corollary 1.2 and 2.2 and their optimal selections with respect to the order of $m$.

$n$ represents the "topology" in CFL which indicates the connections and $\kappa_\lambda$ represents the impacts from the topology in DFL. Therefore, to understand which mode is better, we can directly study the impacts of their connections. In Table 4, we could know `FedAvg` always generalizes better than `DFedAvg`, which largely benefits from regularly averaging on a global server. In the whole training process, centralized approaches always maintain a high global consensus. Though Lian et al. (2017) have learned that the computational complexity of the C/D-PSGD may be similar in the optimization, from the perspective of test error and excess risk, CFL shows stronger stability

Table 4: Comparison of the order of generalization $\varepsilon_G$ on $n$ in CFL and $\kappa_\lambda$ in DFL.

| | FedAvg | DFedAvg |
|---|---|---|
| $\varepsilon_G$ | $\mathcal{O}\left(\frac{1}{S}\left(\frac{n^{\frac{\mu L}{1+\mu L}}}{m}\right)\right)$ | $\mathcal{O}\left(\frac{1}{S}\left(\frac{1+6\sqrt{m}\kappa_\lambda}{m}\right)^{\frac{1}{1+\mu L}}\right)$ |
| best selection to minimize $\varepsilon_G$ | $n=1$ | $\kappa_\lambda=0$ (`full`-topology) |
| best $\varepsilon_G$ | $\mathcal{O}\left(\frac{1}{Sm}\right)$ | $\mathcal{O}\left(\frac{1}{S}\left(\frac{1}{m}\right)^{\frac{1}{1+\mu L}}\right)$ |
| worst selection to maximize $\varepsilon_G$ | $n=m$ (`full`-participation) | $\kappa_\lambda\to\infty$ |
| worst $\varepsilon_G$ | $\mathcal{O}\left(\frac{1}{S}\left(\frac{1}{m}\right)^{\frac{1}{1+\mu L}}\right)$ | $\infty$ |
| best selection to minimize $\varepsilon_G+\varepsilon_O$ | $n=\mathcal{O}\left(m^{\frac{1+\mu L}{1+2\mu L}}\right)<m$ | $\kappa_\lambda=0$ (`full`-topology) |
| corresponding $\varepsilon_G$ | $\mathcal{O}\left(\frac{1}{S}\left(\frac{1}{m}\right)^{\frac{1+\mu L}{1+2\mu L}}\right)$ | $\mathcal{O}\left(\frac{1}{S}\left(\frac{1}{m}\right)^{\frac{1}{1+\mu L}}\right)$ |

and more excellent generalization ability. However, the high communication costs in centralized approaches nevertheless are unavoidable. The communication bottleneck is one of the important concerns restricting the development of federated learning. To achieve reliable performance, the number of active clients $n$ in CFL must satisfy at least a polynomial order of $m$. Too small $n$ will always hurt the performance of the optimization. In the DFL framework, communication is determined by the average degree of the adjacent matrix. For instance, in the exponential topology, the communication achieves $\mathcal{O}(\log m)$ at most in one client (Ying et al., 2021), which is much less than the communication overhead in CFL. Therefore, when we have to consider the communication bottleneck, it is also possible to select DFL at the expense of generalization performance.

We summarize our analysis as follows. We can simply assume that: (1) the communication bandwidth of the global server in CFL is $\rho\times$ wider than the common clients; (2) each local client can support at most $N_D$ connections simultaneously. If $m\le(\rho N_D)^{\frac{1+2\mu L}{1+\mu L}}\le(\rho N_D)^{1.5}$, the centralized approaches definitely performs better. When $m$ is very large, although DFL can save communications as a compromise, its generalization performance will be far worse than CFL.

## 5 EXPERIMENTS

In this part, we mainly introduce the empirical studies including setups, hyperparameter selections, and main experiments to validate our analysis. Other details can be referred to in the Appendix A.

**Dataset and Models.** We mainly test the experiments on the image classification task on the CIFAR-10 dataset (Krizhevsky et al., 2009). Our experiments focus on the validation of the theoretical analysis above and we follow Hsu et al. (2019) to split the dataset with a Dirichlet distribution, which is widely used in the field of federated learning. We denote Dirichlet-$\beta$ as different heterogeneous levels that are controlled by the concentration parameter $\beta=0.1$. The number of total clients is selected from $[100,200,500]$. We adopt the ResNet-18 (He et al., 2016) model as the backbone which is implemented in the Pytorch Model Zoo and follow the previous work (Hsieh et al., 2020) to replace the BatchNorm layers with GroupNorm layers. Details can be referred to in the Appendix A.1.

**Hyperparameters.** We follow the previous studies (Xu et al., 2021; Shi et al., 2023) to set the local learning rate as $0.1$ in both CFL and DFL, which is decayed by $0.998$ per round. Total communication rounds are set as $T=1000$. Weight decay is set as $0.001$ without momentum. To eliminate biases caused by batchsize, we grid search for the corresponding optimal value in each setup from $[10,20,40,80]$. Local epochs are selected from $[5,20]$. Due to the page limitations, other details can be referred to the Appendix A.1 including the curves and performance of their best selections.

### 5.1 CENTRALIZED FEDERATED LEARNING

**Different Participation Ratios.** Our theoretical analysis in Theorem 1 indicates that active ratios in CFL balance the optimization and generalization errors. To achieve the best performance, it usually does not require all clients to participate in the training process per round. As shown in Figure 1, under the total 100 clients and local epochs $E=5$, when it achieves the best performance the active ratio is approximately $30\%$. As the number of total clients increases to $500$, the best selection of the active ratio is approximately $40\%$. When this critical value is exceeded, continuing to increase the

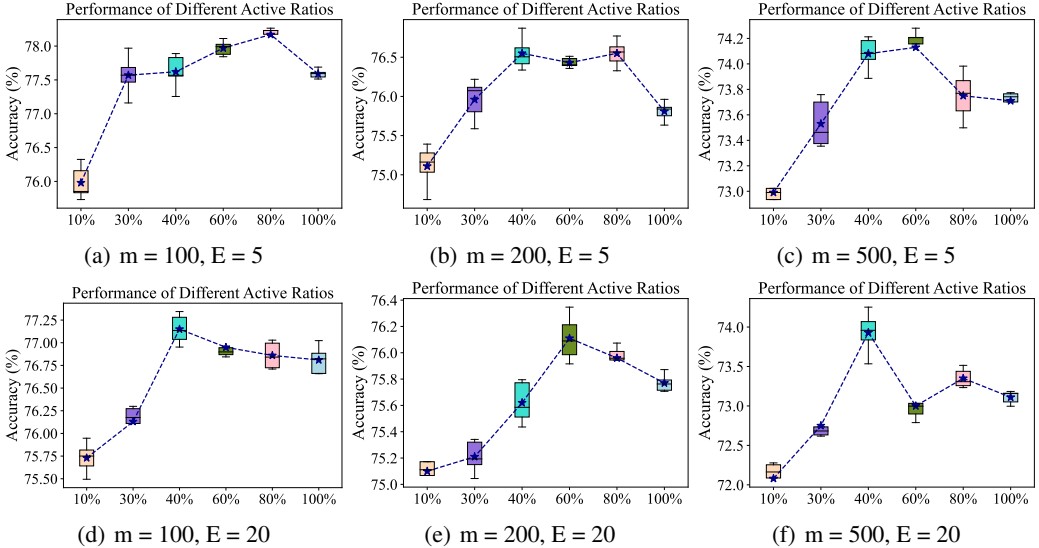

Figure 1: We test different active ratios in CFL on the CIFAR-10 dataset with the ResNet-18 model. $m$ is the number of clients and $E$ is the number of local epochs. Each setup is repeated 5 times.

Table 5: Comparison on the CFL and DFL. In the CFL setup, we assume that the global server can support $2\times$ the communication capabilities of local clients. `FedAvg-n` means the number of active clients equals $n$, which is determined by the corresponding topology. Each setup is repeated 3 times.

| | $m = 100$ | | $m = 200$ | | $m = 500$ | |
|---|---|---|---|---|---|---|
| | $E = 5$ | $E = 20$ | $E = 5$ | $E = 20$ | $E = 5$ | $E = 20$ |
| DFEDAVG-RING | $59.20_{\pm.15}$ | $46.65_{\pm.17}$ | $50.14_{\pm.17}$ | $50.70_{\pm.12}$ | $41.19_{\pm.14}$ | $45.20_{\pm.11}$ |
| FEDAVG-3 | $67.71_{\pm.48}$ | $68.96_{\pm.26}$ | $67.06_{\pm.33}$ | $66.66_{\pm.41}$ | $63.33_{\pm.24}$ | $62.72_{\pm.47}$ |
| FEDAVG-6 | $74.17_{\pm.35}$ | $73.63_{\pm.39}$ | $73.08_{\pm.31}$ | $72.00_{\pm.27}$ | $68.05_{\pm.54}$ | $67.50_{\pm.34}$ |
| DFEDAVG-GRID | $73.27_{\pm.19}$ | $73.45_{\pm.11}$ | $67.70_{\pm.09}$ | $68.60_{\pm.17}$ | $58.20_{\pm.39}$ | $59.36_{\pm.17}$ |
| FEDAVG-5 | $73.13_{\pm.37}$ | $72.97_{\pm.33}$ | $71.95_{\pm.26}$ | $70.84_{\pm.35}$ | $67.10_{\pm.38}$ | $66.27_{\pm.55}$ |
| FEDAVG-10 | $75.48_{\pm.44}$ | $75.03_{\pm.29}$ | $73.97_{\pm.18}$ | $73.39_{\pm.57}$ | $71.06_{\pm.27}$ | $70.44_{\pm.41}$ |
| DFEDAVG-EXP | $76.54_{\pm.11}$ | $\mathbf{76.12_{\pm.08}}$ | $74.05_{\pm.12}$ | $74.43_{\pm.15}$ | $67.28_{\pm.12}$ | $68.11_{\pm.13}$ |
| FEDAVG-$\log m$ | $75.26_{\pm.42}$ | $74.42_{\pm.36}$ | $74.11_{\pm.29}$ | $73.93_{\pm.22}$ | $70.04_{\pm.28}$ | $69.55_{\pm.34}$ |
| FEDAVG-$2\log m$ | $\mathbf{77.19_{\pm.17}}$ | $76.09_{\pm.23}$ | $\mathbf{75.57_{\pm.26}}$ | $\mathbf{74.86_{\pm.17}}$ | $\mathbf{71.61_{\pm.34}}$ | $\mathbf{71.49_{\pm.28}}$ |

active ratio causes significant performance degradation. The optimal active ratio is roughly between 40% and 80%. When all clients participating in the training as $n = m$, CFL could be considered as DFL on the full-connected topology. Therefore, we can intuitively see the poor generalization of the decentralized in Figure 1, i.e., full participation is worse than partial participation.

**Different Local Intervals.** According to the Corollary 1.2 and the corresponding discussions, we know the optimal number of the active ratio will decrease as the local interval $K$ increases. Figure 1 also validates this in the experiments. When we fix the total clients $m$, we can see that the best performance corresponds to smaller active ratios when local epochs $E$ increase from 5 to 20. For instance, when $m = 100$, the optimal selection of the active ratio approximately decreases from 80% to 40%. And, when the local interval $K$ increases, we can see that test errors increase. As claimed by Karimireddy et al. (2020), FL suffers from the client-drift problem. When $K$ is large enough, local models will overfit the local optimums and get far away from the global optimum. Due to the space limitation, full curves of the loss and accuracy are stated in Appendix A.2.1.

## 5.2 DECENTRALIZED FEDERATED LEARNING

**CFL v.s. DFL.** Table 5 shows the test accuracy between DFL with different topologies and corresponding CFL. To achieve fair comparisons, we copy the DFL's optimal hyperparameters to the CFL setup and select a proper partial participation ratio in CFL which follows that the communication bandwidth of the global server is $1\times$ or $2\times$ of the local clients. For instance, ring-topology is

equivalent to 3 local clients jointly train one model. Therefore, we test the number of active clients in CFL equals 3 ($1\times$) and 6 ($2\times$) respectively. In fact, the global server is much more than twice the communication capacity of local devices in practice. From the results, we can clearly see CFL always generalizes better than the DFL on $2\times$ bandwidth. Even if the global server has the same bandwidth as local clients, DFL is only slightly better than CFL on the $m = 100$ setup. With the increase of $m$, DFL's performance degradation is very severe. This is also in line with our conclusions in Table 1 and 4, which demonstrates the poor generalization and excess risk in DFL. Actually, it doesn't save as much bandwidth as one might think in practice, especially with high heterogeneity. When $m$ is large enough, even though CFL and DFL maintain similar communication costs, the generalization performance of CFL is much higher than that of DFL.

**Performance Collapse.** Another important point of our analysis is the potential performance collapse in DFL. In Table 5, because the total amount of data remains unchanged, the number of local data samples $S$ will decrease as $m$ increases. To eliminate this impact, we fix the local amount of data $S = 100$ for each client in the horizontal comparison to validate the minimal condition required to avoid performance collapse. $m \times S$ means the total data samples. Under a fixed $S$, increasing $m$ also means enlarging the dataset.

Table 6: Performance collapse in DFL if the topology does not satisfy the minimal condition. We fix $S = 100$ and increase the $m$ to test performance trends (arrow) on common topologies.

| total $m$ | 300 | 350 | 400 | 450 | 500 |
|---|---|---|---|---|---|
| full | 67.86 | 69.90 | 70.94 | 72.10 | **73.12** ↑ |
| exp | 64.64 | 65.85 | 66.23 | 67.04 | **67.28** ↑ |
| grid | 57.28 | 58.14 | **59.11** | 58.31 | 58.20 ↓ |
| ring | 43.68 | **44.54** | 43.56 | 42.01 | 41.19 ↓ |

As shown in Table 6, we can clearly see that on the full and exp topologies, increasing clients can effectively increase the test accuracy. However, on the grid and ring topologies, since they do not satisfy the minimal condition of $\kappa_\lambda$ as shown in Table 3, even increasing $m$ (enlarging dataset) will cause significant performance degradation.

## 6 CONCLUSION

In this paper, we provide the analysis of the uniform stability and excess risk between CFL and DFL without the idealized assumption of bounded gradients. Our analysis provides a comprehensive and novel understanding of the comparison between CFL and DFL. From the generalization perspective, CFL is always better than DFL. Furthermore, we prove that to achieve minimal excess risk and test error, CFL only requires partial local clients to participate in the training per round. Moreover, though decentralized approaches are adopted as the compromise of centralized ones which could significantly reduce the communication rounds theoretically, the topology must satisfy the minimal requirement to avoid performance collapse. In summary, our analysis clearly answers the *question* in Section 1 and points out how to choose the suitable training mode in real-world scenarios.

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

In the appendix, we state additional experiments (Section A) and introduce detailed proofs of the main theorems (Section B).

# A    ADDITIONAL EXPERIMENTS

In this part, we mainly introduce the details of experimental setups and some additional experimental results to validate the conclusions of the theoretical analysis. To make the curves smooth, we use the `tsmoothie.smoother` package and the `ConvolutionSmoother` function to adjust the curves (`window_len=30, window_type='hanning'`).

## A.1    SETUPS DETAILS

**Dataset.**    The CIFAR dataset (Krizhevsky et al., 2009) is a fundamental dataset in the computer version tasks. We use the CIFAR-10 dataset. There are 50,000 images for training and 10,000 images for testing. Each data sample is with a resolution of $3 \times 32 \times 32$.

Table 7: Dataset introductions.

|  | Training | Testing | Categories | Resolution |
|---|---|---|---|---|
| CIFAR-10 | 50,000 | 10,000 | 10 | $3\times32\times32$ |

**Model.**    ResNet (He et al., 2016) is a fundamental backbone in the studies of federated learning. Most previous works have performed validation experiments based on this model. However, there are various fine-tuning structures adopted in different studies which makes the test accuracy claimed in different papers difficult to compare directly. We also try some structural modifications, i.e. using a small convolution size at the beginning, which may improve the performance without any other tricks. In order to avoid errors in reproduction or comparison, we use the implementation in the Pytorch Model Zoo without other handcraft adjustments. The out dimension of the last linear layer is decided by the total classes of the dataset.

**Hyperparameter Selections.**    We select different hyperparameters to ensure the models can be well-trained fairly on the two setups. Details of the selections are shown as follows.

- **Batchsize:** Current works mainly prefer two selections of the batchsize. One is 50 (Karim-ireddy et al., 2020; Acar et al., 2021; Sun et al., 2023b;a), and the other is 128 (Qu et al., 2022; Shi et al., 2023). Xu et al. (2021) also discuss some different selections based on manual adjustments. Motivated by the previous works and the fact that batchsize is affected by the local intervals and the learning rate, we conduct extensive experiments and select the optimal value which could achieve significant performance.

|  | Total Clients | E | Selection | Optimal |
|---|---|---|---|---|
| CFL | m = 500 | 5 |  | 10 |
|  | m = 200 |  |  | 20 |
|  | m = 100 |  | $[10, 20, 40, 80]$ | 20 |
|  | m = 500 | 20 |  | 20 |
|  | m = 200 |  |  | 40 |
|  | m = 100 |  |  | 80 |
| DFL | m = 500 | 5 |  | 40 |
|  | m = 200 |  |  | 40 |
|  | m = 100 |  | $[10, 20, 40, 80]$ | 40 |
|  | m = 500 | 20 |  | 40 |
|  | m = 200 |  |  | 80 |
|  | m = 100 |  |  | 80 |

Corresponding curves are shown in Figure 2.

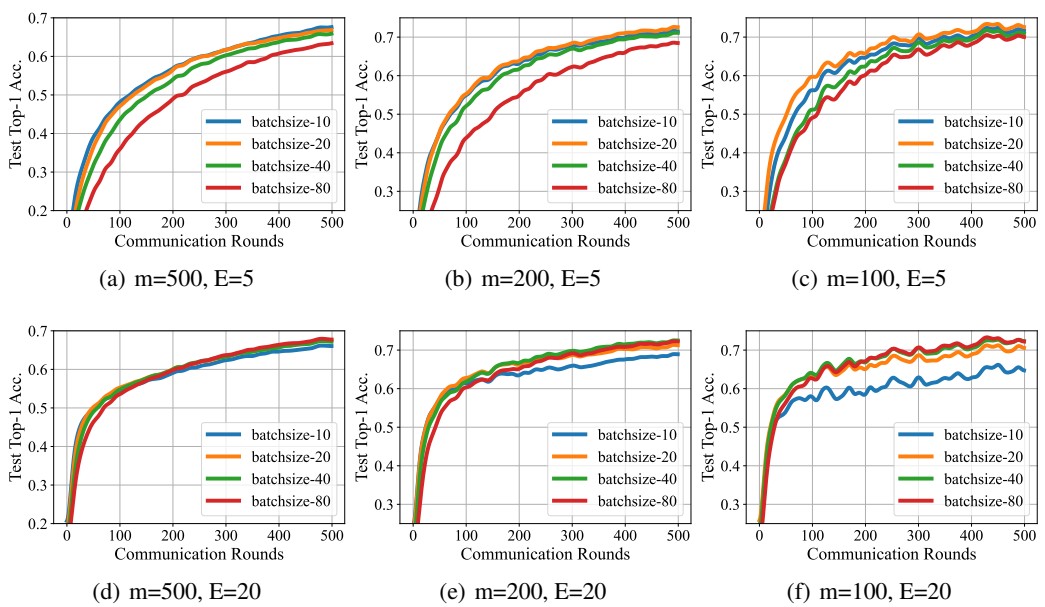

Figure 2: We test different batchsizes in **CFL** on the Dirichlet-0.1 split of the CIFAR-10 dataset with the ResNet-18 models. $m$ is the number of clients and $E$ is the number of local epochs.

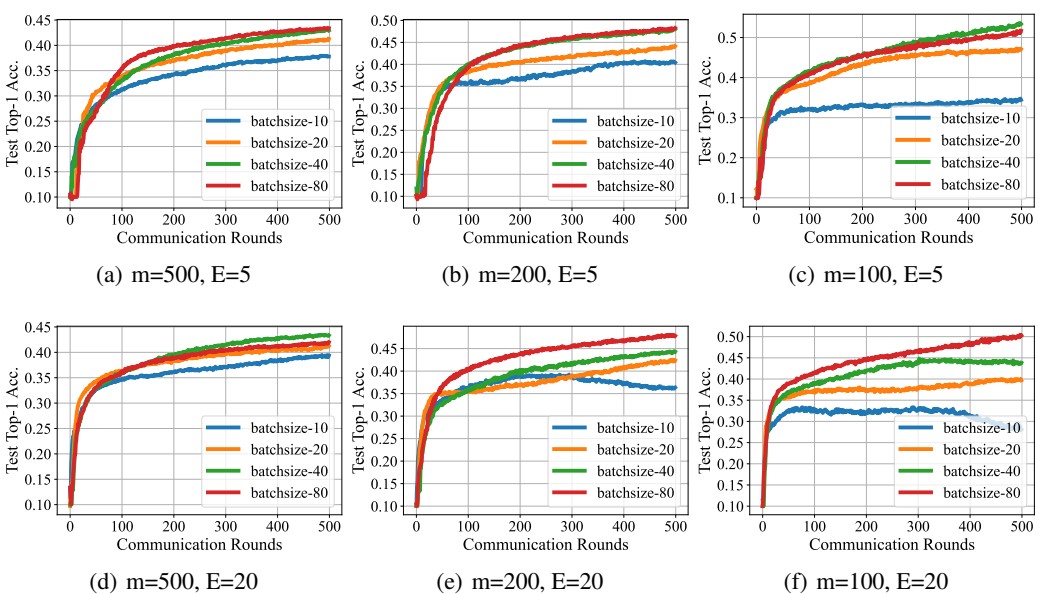

Figure 3: We test different batchsizes in **DFL** on the Dirichlet-0.1 split of the CIFAR-10 dataset with the ResNet-18 models. $m$ is the number of clients and $E$ is the number of local epochs.

- **Learning rate:** In the previous studies, they unanimously select the local learning rate as $0.1$. Xu et al. (2021); Shi et al. (2023) also test different selections and confirm the optimal selection as $0.1$. We follow this selection to fairly compare their performance.

- **Weight Decay:** We test some common selections from $[0.01, 0.001, 0.0005, 0.0001]$. Its optimal selection jumps between $0.01$ and $0.005$. Even in similar scenarios, there will be some differences in their optimal values. The fluctuation amplitude reflected in the test accuracy is about $0.4\%$. For a fair comparison, we fix it as a median value of $0.001$.

## A.2 ADDITIONAL EXPERIMENTS

### A.2.1 DIFFERENT ACTIVE RATIOS IN CFL

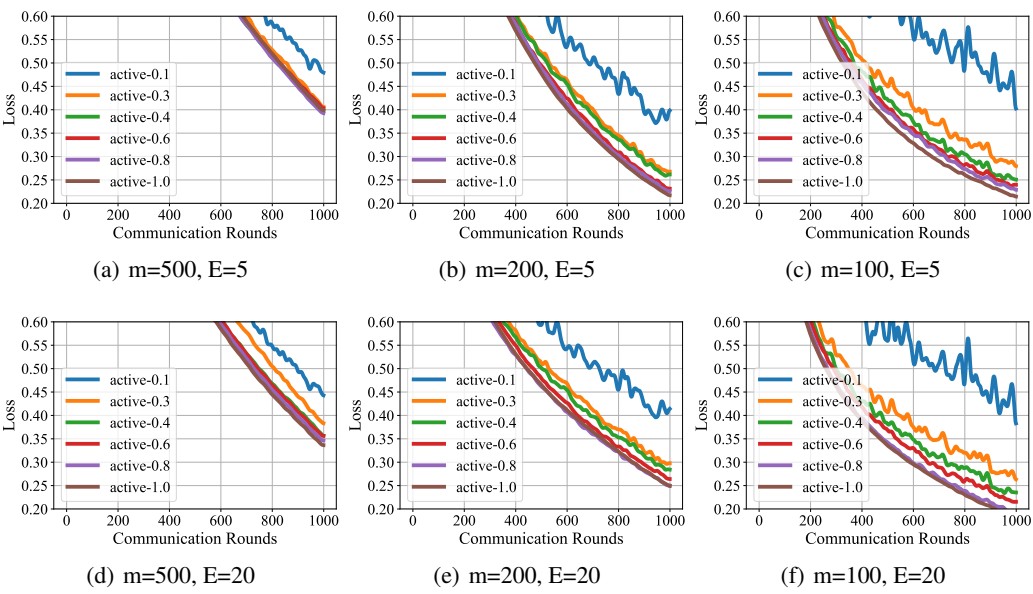

Figure 4: **Loss** curves of different active ratios in CFL.

Obviously, increasing $n$ helps to accelerate the optimization. A larger active ratio means a faster convergence rate. As shown in Figure 4, we can see this phenomenon very clearly in the subfigure (c) and (f). Though the real acceleration is not as fast as linear speedup, from the optimization perspective, increasing the active ratio can truly achieve a lower loss value. This is also consistent with the conclusions of previous work in the optimization process analysis.

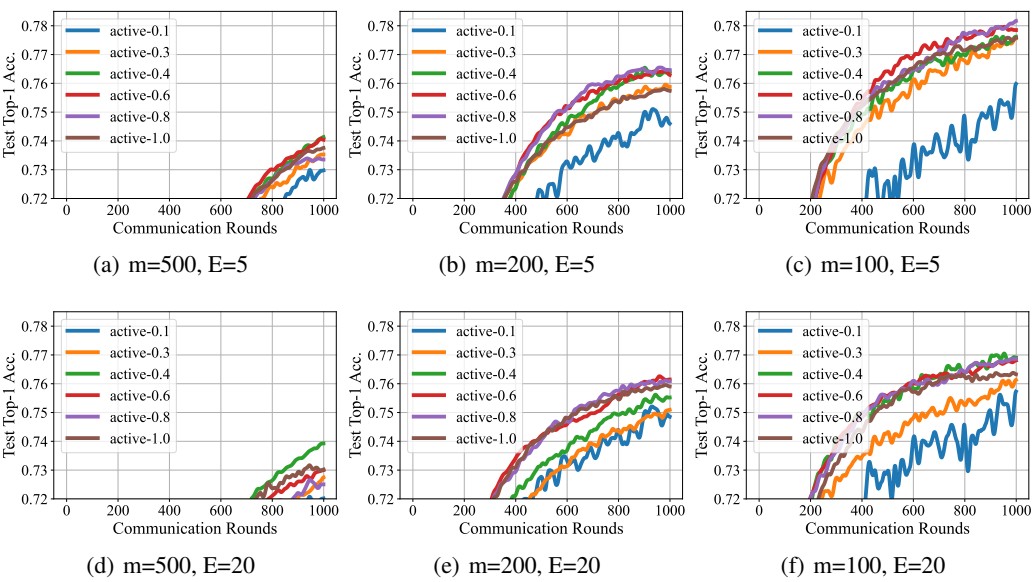

Figure 5: **Accuracy** curves of different active ratios in CFL.

However, as shown in Figure 5, increasing $n$ does not always mean higher test accuracy. In CFL, there is an optimal active ratio, which means the active number of clients is limited. In our analysis, the optimal selection is between the order of $m^{\frac{2}{3}}$ and $m$. In practice, it is about from $0.4m$ to $0.8m$.

### A.2.2 DIFFERENT TOPOLOGY IN DFL

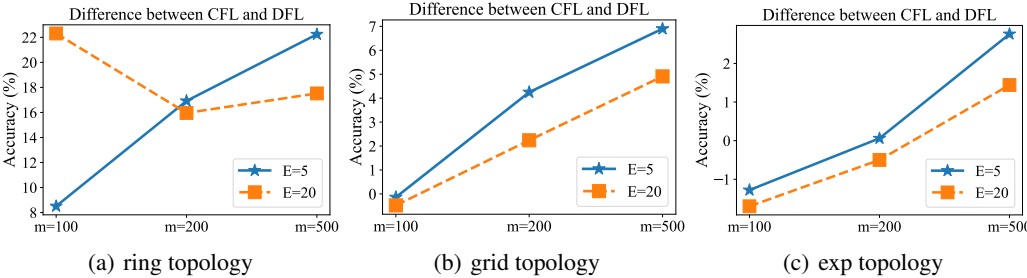

Figure 6: **Accuracy** curves of different topologies in DFL and corresponding CFL.

As Figure 6 shows, under similar communication costs, DFL can not be better than CFL. FL always maintains stronger generalization performance, except at very low active ratios where its performance is severely compromised. We can also observe a more significant phenomenon. The gap between CFL and DFL will be larger as $m$ increases. This is also lined with our analysis.

Figure 7: **Accuracy** difference of different topologies between DFL and corresponding CFL.

As shown in Figure 7, we can clearly see that the difference between CFL and DFL increases significantly as $m$ increases. We calculate the difference by (*test accuracy* of CFL - *test accuracy* of DFL) under the same communication costs. According to our analysis, the excess risk of CFL is much smaller than that of DFL, which is $\mathcal{O}(m^{-\frac{1+\mu L}{1+2\mu L}})$ and $\mathcal{O}(m^{-\frac{1}{1+\mu L}})$ respectively. Therefore, we can know that CFL generalizes better than DFL at most with the rate of $\mathcal{O}(m^{\frac{(\mu L)^2}{(1+\mu L)(1+2\mu L)}})$. In general cases, the worst generalization of CFL equals to the best generalization of DFL. Therefore, when the generalization error dominates the test accuracy, CFL is always better than DFL.

# B  PROOF OF THEOREMS AND LEMMAS

In this part, we mainly introduce the proofs of the theorems and some important lemmas which are adopted in this work.

## B.1  IMPORTANT LEMMAS

Both centralized and decentralized FL setups minimize the finite-sum problem. Therefore, we denote $\Delta_k^t = \sum_{i \in [m]} \|w_{i,k}^t - \widetilde{w}_{i,k}^t\|$ as the average difference. Here we define an event $\xi$. If $\Delta_{k_0}^{t_0} = 0$ happens at $k_0$-th iteration of $t_0$-th round, $\xi = 1$ otherwise 0. Because the index of the data samples on the joint dataset $\mathcal{C}$ and $\widetilde{\mathcal{C}}$ are selected simultaneously, it describes whether the different samples in the two datasets have been selected before $k_0$-th iteration of $t_0$-th round. We denote $\tau = tK + k$ as the index of training iterations. We denote $z_{i^\star, j^\star}$ and $\widetilde{z}_{i^\star, j^\star}$ are the only different data samples between the dataset $\mathcal{C}$ and $\widetilde{\mathcal{C}}$. The following Table 8 summarizes the details.

Table 8: Additional notations adopted in the proofs.

| Notation | Formulation | Description |
|---|---|---|
| $\tau$ | $tK + k$ | index of the iterations |
| $\tau_0$ | $t_0 K + k_0$ | index of the observed iteration in event $\xi$ |
| $(i^\star, j^\star)$ | - | index of the different data sample on $\mathcal{C}$ and $\widetilde{\mathcal{C}}$ |
| $\Delta_k^t$ | $\sum_{i \in [m]} \|w_{i,k}^t - \widetilde{w}_{i,k}^t\|$ | local stability on $k$-th iteration of $t$-th round |

**Lemma 4 (Stability in `FedAvg`)** *Let function $f(w)$ satisfies Assumption 3, the models $w^T = \mathcal{A}(\mathcal{C})$ and $\widetilde{w}^T = \mathcal{A}(\widetilde{\mathcal{C}})$ are generated after $T$ training rounds by the centralized `FedAvg` method (Algorithm 1), we can bound their objective difference as:*

$$\mathbb{E}\left[|f(w^T; z) - f(\widetilde{w}^T; z)|\right] \le G\mathbb{E}\left[\|w^T - \widetilde{w}^T\| \mid \xi\right] + \frac{nU\tau_0}{mS}. \tag{12}$$

*where $U = \sup_{w,z} f(w; z) < +\infty$ is the upper bound of the loss and $\tau_0 = t_0 K + k_0$ is a specific index of the total iterations.*

**Proof.**  Via the expansion of the probability we have:

$$\mathbb{E}\left[|f(w^T; z) - f(\widetilde{w}^T; z)|\right]$$
$$= P(\xi)\mathbb{E}\left[|f(w^T; z) - f(\widetilde{w}^T; z)| \mid \xi\right] + P(\xi^c)\mathbb{E}\left[|f(w^T; z) - f(\widetilde{w}^T; z)| \mid \xi^c\right]$$
$$\le G\mathbb{E}\left[\|w^T - \widetilde{w}^T\| \mid \xi\right] + UP(\xi^c).$$

Let the variable $I$ assume the index of the first time to use the data sample $\widetilde{z}_{i^\star, j^\star}$ on the dataset $\widetilde{\mathcal{S}}_{i^\star}$. When $I > t_0 K + k_0$, then $\Delta_{k_0}^{t_0} = 0$ must happens. Thus we have:

$$P(\xi^c) = P(\Delta_{k_0}^{t_0} > 0) \le P(I \le t_0 K + k_0). \tag{13}$$

On each step $\tau$, the data sample is uniformly sampled from the local dataset. When the dataset $\mathcal{S}_{i^\star}$ is selected, the probability of sampling $z_{i^\star, j^\star}$ is $1/S$. Let $\chi$ denote the event that $\mathcal{S}_{i^\star}$ is selected or not. Thus we have the union bound:

$$P(I \le t_0 K + k_0) \le \sum_{t=0}^{t_0-1}\sum_{k=0}^{K-1} P(I = tK + k; \chi) + \sum_{k=0}^{k_0} P(I = t_0 K + k; \chi)$$

$$= \sum_{t=0}^{t_0-1}\sum_{k=0}^{K-1}\sum_{\chi} P(I = tK + k|\chi)P(\chi) + \sum_{k=0}^{k_0}\sum_{\chi} P(I = t_0 K + k|\chi)P(\chi)$$

$$= \frac{n}{m}\left(\sum_{t=0}^{t_0-1}\sum_{k=0}^{K-1} P(I = tK + k) + \sum_{k=0}^{k_0} P(I = t_0 K + k)\right)$$

$$= \frac{n(t_0 K + k_0)}{mS} = \frac{n\tau_0}{mS}.$$

The second equality adopts the fact of random active clients with the probability of $n/m$.  $\square$

**Lemma 5 (Stability in `DFedAvg`)** *Let function $f(w)$ satisfies Assumption 3, the models $w^T = \mathcal{A}(\mathcal{C})$ and $\widetilde{w}^T = \mathcal{A}(\widetilde{\mathcal{C}})$ are generated after $T$ training rounds by the centralized `DFedAvg` method (Algorithm 2), we can bound their objective difference as:*

$$\mathbb{E}\left[|f(w^T; z) - f(\widetilde{w}^T; z)|\right] \le G\mathbb{E}\left[\|w^T - \widetilde{w}^T\| \mid \xi\right] + \frac{U\tau_0}{S}. \tag{14}$$

**Proof.** The most part is the same as the proof in Lemma 4 except the probability $P(\chi) = 1$ in a decentralized federated learning setup (because all clients will participate in the training). $\square$

**Lemma 6 (Same Sample)** *Let the function $f_i$ satisfies Assumption 1, and the local updates be $w_{i,k+1}^t = w_{i,k}^t - \eta g_{i,k}^t$ and $\widetilde{w}_{i,k+1}^t = \widetilde{w}_{i,k}^t - \eta\widetilde{g}_{i,k}^t$, by sampling the same data $z$ (not the $z_{i^\star, j^\star}$), we have:*

$$\mathbb{E}\|w_{i,k+1}^t - \widetilde{w}_{i,k+1}^t\| \le (1 + \eta L)\mathbb{E}\|w_{i,k}^t - \widetilde{w}_{i,k}^t\|. \tag{15}$$

**Proof.** In each round $t$, we have:

$$\begin{aligned}
\mathbb{E}\|w_{i,k+1}^t - \widetilde{w}_{i,k+1}^t\| &= \mathbb{E}\|w_{i,k}^t - \widetilde{w}_{i,k}^t - \eta(g_{i,k}^t - \widetilde{g}_{i,k}^t)\| \\
&\le \mathbb{E}\|w_{i,k}^t - \widetilde{w}_{i,k}^t\| + \eta\mathbb{E}\|\nabla f_i(w_{i,k}^t, z) - \nabla f_i(\widetilde{w}_{i,k}^t, z)\| \\
&\le (1 + \eta L)\mathbb{E}\|w_{i,k}^t - \widetilde{w}_{i,k}^t\|.
\end{aligned}$$

$\square$

**Lemma 7 (Different Sample)** *Let the function $f_i$ satisfies Assumption 1 and 2, and the local updates be $w_{i^\star, k+1}^t = w_{i^\star, k}^t - \eta g_{i^\star, k}^t$ and $\widetilde{w}_{i^\star, k+1}^t = \widetilde{w}_{i^\star, k}^t - \eta\widetilde{g}_{i^\star, k}^t$, by sampling the different data samples $z_{i^\star, j^\star}$ and $\widetilde{z}_{i^\star, j^\star}$ (simplified to $z$ and $\widetilde{z}$), we have:*

$$\mathbb{E}\|w_{i^\star, k+1}^t - \widetilde{w}_{i^\star, k+1}^t\| \le (1 + \eta L)\mathbb{E}\|w_{i^\star, k}^t - \widetilde{w}_{i^\star, k}^t\| + 2\eta\sigma_l. \tag{16}$$

**Proof.** In each round $t$, we have:

$$\begin{aligned}
\mathbb{E}\|w_{i^\star, k+1}^t - \widetilde{w}_{i^\star, k+1}^t\| &= \mathbb{E}\|w_{i^\star, k}^t - \widetilde{w}_{i^\star, k}^t - \eta(g_{i^\star, k}^t - \widetilde{g}_{i^\star, k}^t)\| \\
&\le \mathbb{E}\|w_{i^\star, k}^t - \widetilde{w}_{i^\star, k}^t\| + \eta\mathbb{E}\|\nabla f_{i^\star}(w_{i^\star, k}^t, z) - \nabla f_{i^\star}(\widetilde{w}_{i^\star, k}^t, \widetilde{z})\| \\
&\le \mathbb{E}\|w_{i^\star, k}^t - \widetilde{w}_{i^\star, k}^t\| + \eta\mathbb{E}\|\nabla f_{i^\star}(w_{i^\star, k}^t, z) - \nabla f_{i^\star}(\widetilde{w}_{i^\star, k}^t, z)\| \\
&\quad + \eta\mathbb{E}\|\nabla f_{i^\star}(\widetilde{w}_{i^\star, k}^t, z) - \nabla f_{i^\star}(\widetilde{w}_{i^\star, k}^t, \widetilde{z})\| \\
&\le (1 + \eta L)\mathbb{E}\|w_{i^\star, k}^t - \widetilde{w}_{i^\star, k}^t\| + \eta\mathbb{E}\|\nabla f_{i^\star}(\widetilde{w}_{i^\star, k}^t, z) - \nabla f_{i^\star}(\widetilde{w}_{i^\star, k}^t)\| \\
&\quad + \eta\mathbb{E}\|\nabla f_{i^\star}(\widetilde{w}_{i^\star, k}^t, \widetilde{z}) - \nabla f_{i^\star}(\widetilde{w}_{i^\star, k}^t)\| \\
&\le (1 + \eta L)\mathbb{E}\|w_{i^\star, k}^t - \widetilde{w}_{i^\star, k}^t\| + 2\eta\sigma_l.
\end{aligned}$$

The last inequality adopts $\mathbb{E}[x] = \sqrt{(\mathbb{E}[x])^2} = \sqrt{\mathbb{E}[x^2] - \mathbb{E}[x - \mathbb{E}[x]]^2} \le \sqrt{\mathbb{E}[x^2]}$. $\square$

**Lemma 8 (Upper Bound of Aggregation Gaps)** *According to Algorithm 1 and 2, the aggregation of centralized federated learning is $w_{i,0}^{t+1} = w^{t+1} = \frac{1}{n}\sum_{\mathcal{N}} w_{i,K}^t$, and the aggregation of decentralized federated learning is $w_{i,0}^{t+1} = \sum_{j\in\mathcal{A}_i} a_{ij}w_{i,K}^t$. On both setups, we can upper bound the aggregation gaps by:*

$$\Delta_0^{t+1} \le \Delta_K^t. \tag{17}$$

**Proof.** We prove them respectively.

(1) Centralized federated learning setup (Acar et al., 2021).

In centralized federated learning, we select a subset $\mathcal{N}$ in each communication round. Thus we have:

$$\begin{aligned}
&\Delta_0^{t+1} \\
&= \sum_{i\in[m]}\mathbb{E}\|w_{i,0}^{t+1} - \widetilde{w}_{i,0}^{t+1}\| = \sum_{i\in[m]}\mathbb{E}\|w^{t+1} - \widetilde{w}^{t+1}\| = \sum_{i\in[m]}\mathbb{E}\|\frac{1}{n}\sum_{i\in\mathcal{N}}\left(w_{i,K}^t - \widetilde{w}_{i,K}^t\right)\|
\end{aligned}$$

$$\leq \sum_{i\in[m]} \frac{1}{n}\mathbb{E}\left[\sum_{i\in\mathcal{N}} \|w_{i,K}^t - \widetilde{w}_{i,K}^t\|\right] = \sum_{i\in[m]} \frac{1}{n}\frac{n}{m}\sum_{i\in[m]} \mathbb{E}\|w_{i,K}^t - \widetilde{w}_{i,K}^t\|$$

$$= \sum_{i\in[m]} \frac{1}{m}\sum_{i\in[m]} \mathbb{E}\|w_{i,K}^t - \widetilde{w}_{i,K}^t\| = \sum_{i\in[m]} \mathbb{E}\|w_{i,K}^t - \widetilde{w}_{i,K}^t\| = \Delta_K^t.$$

(2) Decentralized federated learning setup.

In decentralized federated learning, we aggregate the models in each neighborhood. Thus we have:

$$\Delta_0^{t+1}$$
$$= \sum_{i\in[m]} \mathbb{E}\|w_{i,0}^{t+1} - \widetilde{w}_{i,0}^{t+1}\| = \sum_{i\in[m]} \mathbb{E}\|\sum_{j\in\mathcal{A}_i} a_{ij}\left(w_{j,K}^t - \widetilde{w}_{j,K}^t\right)\| \leq \sum_{i\in[m]}\sum_{j\in\mathcal{A}_i} a_{ij}\mathbb{E}\|w_{j,K}^t - \widetilde{w}_{j,K}^t\|$$
$$= \sum_{j\in[m]}\sum_{i\in\mathcal{A}_j} a_{ji}\mathbb{E}\|w_{j,K}^t - \widetilde{w}_{j,K}^t\| \leq \sum_{j\in[m]} \mathbb{E}\|w_{j,K}^t - \widetilde{w}_{j,K}^t\| = \Delta_K^t.$$

The last equality adopts the symmetry of the adjacent matrix $\mathbf{A} = \mathbf{A}^\top$. □

**Lemma 9 (Recursion)** *According to the Lemma 6 and 7, we can bound the recursion in the local training:*

$$\Delta_{k+1}^t + \frac{2\sigma_l}{SL} \leq (1+\eta L)\left(\Delta_k^t + \frac{2\sigma_l}{SL}\right). \tag{18}$$

**Proof.** In each iteration, the specific $j^\star$-th data sample in the $\mathcal{S}_{i^\star}$ and $\widetilde{\mathcal{S}}_{i^\star}$ is uniformly selected with the probability of $1/S$. In other datasets $\mathcal{S}_i$, all the data samples are the same. Thus we have:

$$\Delta_{k+1}^t = \sum_{i\neq i^\star} \mathbb{E}\left[\|w_{i,k+1}^t - \widetilde{w}_{i,k+1}^t\|\right] + \mathbb{E}\left[\|w_{i^\star,k+1}^t - \widetilde{w}_{i^\star,k+1}^t\|\right]$$

$$\leq (1+\eta L)\sum_{i\neq i^\star} \mathbb{E}\left[\|w_{i,k+1}^t - \widetilde{w}_{i,k+1}^t\|\right] + \left(1 - \frac{1}{S}\right)(1+\eta L)\mathbb{E}\left[\|w_{i^\star,k}^t - \widetilde{w}_{i^\star,k}^t\|\right]$$

$$+ \frac{1}{S}\left[(1+\eta L)\mathbb{E}\left[\|w_{i^\star,k}^t - \widetilde{w}_{i^\star,k}^t\|\right] + 2\eta\sigma_l\right] = (1+\eta L)\Delta_k^t + \frac{2\eta\sigma_l}{S}.$$

There we can bound the recursion formulation as $\Delta_{k+1}^t + \frac{2\sigma_l}{SL} \leq (1+\eta L)\left(\Delta_k^t + \frac{2\sigma_l}{SL}\right).$ □

**Lemma 10** *For $0 < \lambda < 1$ and $0 < \alpha < 1$, we have the following inequality:*

$$\sum_{s=0}^{t-1} \frac{\lambda^{t-s-1}}{(s+1)^\alpha} \leq \frac{\kappa_\lambda}{t^\alpha}, \tag{19}$$

*where* $\kappa_\lambda = \left(\frac{\alpha}{e}\right)^\alpha \frac{1}{\lambda\left(\ln\frac{1}{\lambda}\right)^\alpha} + \frac{2^\alpha}{(1-\alpha)e\lambda\ln\frac{1}{\lambda}} + \frac{2^\alpha}{\lambda\ln\frac{1}{\lambda}}.$

**Proof.** According to the accumulation, we have:

$$\sum_{s=0}^{t-1} \frac{\lambda^{t-s-1}}{(s+1)^\alpha} = \lambda^{t-1} + \sum_{s=1}^{t-1} \frac{\lambda^{t-s-1}}{(s+1)^\alpha} \leq \lambda^{t-1} + \int_{s=1}^{s=t} \frac{\lambda^{t-s-1}}{s^\alpha}ds$$

$$= \lambda^{t-1} + \int_{s=1}^{s=\frac{t}{2}} \frac{\lambda^{t-s-1}}{s^\alpha}ds + \int_{s=\frac{t}{2}}^{s=t} \frac{\lambda^{t-s-1}}{s^\alpha}ds$$

$$\leq \lambda^{t-1} + \lambda^{\frac{t}{2}-1}\int_{s=1}^{s=\frac{t}{2}} \frac{1}{s^\alpha}ds + \left(\frac{2}{t}\right)^\alpha\int_{s=\frac{t}{2}}^{s=t} \lambda^{t-s-1}ds$$

$$\leq \lambda^{t-1} + \lambda^{\frac{t}{2}-1}\frac{1}{1-\alpha}\left(\frac{t}{2}\right)^{1-\alpha} + \left(\frac{2}{t}\right)^\alpha\frac{\lambda^{-1}}{\ln\frac{1}{\lambda}}.$$

Thus we have LHS $\leq \frac{1}{t^\alpha}\left(\lambda^{t-1}t^\alpha + \lambda^{\frac{t}{2}-1}\frac{t}{(1-\alpha)2^{1-\alpha}} + \frac{2^\alpha}{\lambda\ln\frac{1}{\lambda}}\right)$. The first term can be bounded as $\lambda^{t-1}t^\alpha \leq \left(\frac{\alpha}{e}\right)^\alpha\frac{1}{\lambda\left(\ln\frac{1}{\lambda}\right)^\alpha}$ and the second term can be bounded as $\lambda^{\frac{t}{2}-1}t \leq \frac{2}{e\lambda\ln\frac{1}{\lambda}}$, which indicates

the selection of the constant $\kappa_\lambda = \left(\frac{\alpha}{e}\right)^\alpha \frac{1}{\lambda\left(\ln\frac{1}{\lambda}\right)^\alpha} + \frac{2^\alpha}{(1-\alpha)e\lambda\ln\frac{1}{\lambda}} + \frac{2^\alpha}{\lambda\ln\frac{1}{\lambda}}$. Furthermore, if $0 <$ $\alpha \le \frac{1}{2} < 1$, we have $\kappa_\lambda \le \frac{1}{\lambda\left(\ln\frac{1}{\lambda}\right)^\alpha} + \frac{2\sqrt{2}}{e\lambda\ln\frac{1}{\lambda}} + \frac{\sqrt{2}}{\lambda\ln\frac{1}{\lambda}} \le \max\left\{\frac{1}{\lambda}, \frac{1}{\lambda\sqrt{\ln\frac{1}{\lambda}}}\right\} + \frac{(2+e)\sqrt{2}}{e\lambda\ln\frac{1}{\lambda}} =$ $\mathcal{O}\left(\max\left\{\frac{1}{\lambda}, \frac{1}{\lambda\sqrt{\ln\frac{1}{\lambda}}}\right\} + \frac{1}{\lambda\ln\frac{1}{\lambda}}\right)$ with respect to the constant $\lambda$. $\qquad\square$

### B.2 STABILITY OF CENTRALIZED FL (THEOREM 1)

According to the Lemma 8 and 9, it is easy to bound the local stability term. We still obverse it when the event $\xi$ happens, and we have $\Delta_{k_0}^{t_0} = 0$. Therefore, we unwind the recurrence formulation from $T, K$ to $t_0, k_0$. Let the learning rate $\eta = \frac{\mu}{\tau} = \frac{\mu}{tK+k}$ is decayed as the communication round $t$ and iteration $k$ where $\mu \le \frac{1}{L}$ is a specific constant, we have:

$$\Delta_K^T \le \left[\prod_{\tau=(T-1)K+1}^{TK}\left(1 + \frac{\mu L}{\tau}\right)\right]\left(\Delta_0^T + \frac{2\sigma_l}{SL}\right) \le \left[\prod_{\tau=(T-1)K+1}^{TK}\left(1 + \frac{\mu L}{\tau}\right)\right]\left(\Delta_K^{T-1} + \frac{2\sigma_l}{SL}\right)$$

$$\le \left[\prod_{\tau=t_0K+k_0+1}^{TK}\left(1 + \frac{\mu L}{\tau}\right)\right]\left(\Delta_{k_0}^{t_0} + \frac{2\sigma_l}{SL}\right)$$

$$\le \left[\prod_{\tau=t_0K+k_0+1}^{TK} e^{\left(\frac{\mu L}{\tau}\right)}\right]\left(\frac{2\sigma_l}{SL}\right) = e^{\mu L\left(\sum_{\tau=t_0K+k_0+1}^{TK}\frac{1}{\tau}\right)}\frac{2\sigma_l}{SL}$$

$$\le e^{\mu L\ln\left(\frac{TK}{t_0K+k_0}\right)}\frac{2\sigma_l}{SL} \le \left(\frac{TK}{\tau_0}\right)^{\mu L}\frac{2\sigma_l}{SL}.$$

According to the Lemma 4, the first term in the stability (condition is omitted for abbreviation) can be bound as:

$$\mathbb{E}\|w^{T+1} - \widetilde{w}^{T+1}\| = \mathbb{E}\|\frac{1}{n}\sum_{i\in\mathcal{N}}\left(w_{i,K}^T - \widetilde{w}_{i,K}^T\right)\| = \frac{1}{n}\mathbb{E}\|\sum_{i\in\mathcal{N}}\left(w_{i,K}^T - \widetilde{w}_{i,K}^T\right)\|$$

$$\le \frac{1}{n}\mathbb{E}\sum_{i\in\mathcal{N}}\|\left(w_{i,K}^T - \widetilde{w}_{i,K}^T\right)\| = \frac{1}{n}\frac{n}{m}\mathbb{E}\sum_{i\in[m]}\|\left(w_{i,K}^T - \widetilde{w}_{i,K}^T\right)\|$$

$$= \frac{1}{m}\sum_{i\in[m]}\mathbb{E}\|\left(w_{i,K}^T - \widetilde{w}_{i,K}^T\right)\| = \frac{1}{m}\Delta_K^T \le \left(\frac{TK}{\tau_0}\right)^{\mu L}\frac{2\sigma_l}{mSL}.$$

Therefore, we can upper bound the stability in centralized federated learning as:

$$\mathbb{E}|f(w^{T+1}; z) - f(\widetilde{w}^{T+1}; z)| \le G\mathbb{E}\|w^{T+1} - \widetilde{w}^{T+1}\| + \frac{nU\tau_0}{mS} \le \frac{2\sigma_l G}{mSL}\left(\frac{TK}{\tau_0}\right)^{\mu L} + \frac{nU\tau_0}{mS}.$$

Obviously, we can select a proper event $\xi$ with a proper $\tau_0$ to minimize the upper bound. For $\tau \in [1, TK]$, by selecting $\tau_0 = \left(\frac{2\sigma_l G}{nUL}\right)^{\frac{1}{1+\mu L}}(TK)^{\frac{\mu L}{1+\mu L}}$, we can minimize the bound with respect to $\tau_0$ as:

$$\mathbb{E}|f(w^{T+1}; z) - f(\widetilde{w}^{T+1}; z)| \le \frac{2nU\tau_0}{mS} = \frac{2nU}{mS}\left(\frac{2\sigma_l G}{nUL}\right)^{\frac{1}{1+\mu L}}(TK)^{\frac{\mu L}{1+\mu L}}$$

$$\le \frac{4}{S}\left(\frac{\sigma_l G}{L}\right)^{\frac{1}{1+\mu L}}\left(\frac{n^{\frac{\mu L}{1+\mu L}}}{m}\right)(UTK)^{\frac{\mu L}{1+\mu L}}.$$

### B.3 STABILITY OF DECENTRALIZED FL (THEOREM 2)

#### B.3.1 AGGREGATION BOUND WITH SPECTRUM GAPS

The same as the proofs in the last part, according to the Lemma 8 and 9, we also can bound the local stability term. Let the learning rate $\eta = \frac{\mu}{\tau} = \frac{\mu}{tK+k}$ is decayed as the communication round $t$ and iteration $k$ where $\mu$ is a specific constant, we have:

$$\Delta_k^t + \frac{2\sigma_l}{SL} \le \left(\frac{\tau}{\tau_0}\right)^{\mu L}\frac{2\sigma_l}{SL}. \tag{20}$$

If we directly combine this inequality with Lemma 5, we will get the vanilla stability of the vanilla SGD optimizer. However, this will be a larger upper bound which does not help us understand the advantages and disadvantages of decentralization. In the decentralized setups, an important study is learning how to evaluate the impact of the spectrum gaps. Thus we must search for a more precise upper bound than above. Therefore, we calculate a more refined upper bound for its aggregation step (Lemma 8) with the spectrum gap $1 - \lambda$.

Let $\mathbf{W}_k^t = \left[ w_{0,k}^t, w_{1,k}^t, \cdots, w_{m,k}^t \right]^\top$ is the parameter matrix of all clients. In the stability analysis, we focus more on the parameter difference instead. Therefore, we denote the matrix of the parameter differences $\Phi_k^t = \mathbf{W}_k^t - \widetilde{\mathbf{W}}_k^t = \left[ w_{0,k}^t - \widetilde{w}_{0,k}^t, w_{1,k}^t - \widetilde{w}_{1,k}^t, \cdots, w_{m,k}^t - \widetilde{w}_{m,k}^t \right]^\top$ as the difference between the models trained on $\mathcal{C}$ and $\widetilde{\mathcal{C}}$ on the $k$-th iteration of $t$-th communication round. Meanwhile, consider the update rules, we have:

$$\Phi_{k+1}^t = \Phi_k^t - \eta_k^t \Gamma_k^t,$$

where $\Gamma_k^t = \left[ g_{0,k}^t - \widetilde{g}_{0,k}^t, g_{1,k}^t - \widetilde{g}_{1,k}^t, \cdots, g_{m,k}^t - \widetilde{g}_{m,k}^t \right]^\top$.

In the `DFedAvg` method shown in Algorithm 2, the aggregation performs after $K$ local updates which demonstrates that the initial state of each round is $\mathbf{W}_0^t = \mathbf{A}\mathbf{W}_K^{t-1}$. It also works on their difference $\Phi_0^t = \mathbf{A}\Phi_K^{t-1}$. Therefore, we have:

$$\Phi_K^t = \Phi_0^t - \sum_{k=0}^{K-1} \eta_k^t \Gamma_k^t = \mathbf{A}\Phi_K^{t-1} - \sum_{k=0}^{K-1} \eta_k^t \Gamma_k^t.$$

Then we prove the recurrence between adjacent rounds. Let $\mathbf{P} = \frac{1}{m}\mathbf{1}\mathbf{1}^\top \in \mathbb{R}^{m \times m}$ and $\mathbf{I} \in \mathbb{R}^{m \times m}$ is the identity matrix, due to the double stochastic property of the adjacent matrix $\mathbf{A}$, we have:

$$\mathbf{AP} = \mathbf{PA} = \mathbf{P}.$$

Thus we have:

$$(\mathbf{I} - \mathbf{P})\Phi_K^t = (\mathbf{I} - \mathbf{P})\mathbf{A}\Phi_K^{t-1} - (\mathbf{I} - \mathbf{P})\sum_{k=0}^{K-1} \eta_k^t \Gamma_k^t$$

$$= \left( \mathbf{A}\Phi_K^{t-1} - \sum_{k=0}^{K-1} \eta_k^t \Gamma_k^t \right) - \mathbf{PA}\Phi_K^{t-1} + \mathbf{PA}\Phi_K^{t-1} - \mathbf{P}\left( \mathbf{A}\Phi_K^{t-1} - \sum_{k=0}^{K-1} \eta_k^t \Gamma_k^t \right).$$

By taking the expectation of the norm on both sides, we have:

$$\mathbb{E}\| (\mathbf{I} - \mathbf{P})\Phi_K^t \| \leq \mathbb{E}\|\mathbf{A}\Phi_K^{t-1} - \sum_{k=0}^{K-1} \eta_k^t \Gamma_k^t - \mathbf{PA}\Phi_K^{t-1}\| + \mathbb{E}\|\sum_{k=0}^{K-1} \eta_k^t \Gamma_k^t\|$$

$$\leq \mathbb{E}\|\mathbf{A}\Phi_K^{t-1} - \mathbf{PA}\Phi_K^{t-1}\| + 2\mathbb{E}\|\sum_{k=0}^{K-1} \eta_k^t \Gamma_k^t\|$$

$$= \mathbb{E}\| (\mathbf{A} - \mathbf{P})(\mathbf{I} - \mathbf{P})\Phi_K^{t-1}\| + 2\mathbb{E}\|\sum_{k=0}^{K-1} \eta_k^t \Gamma_k^t\|$$

$$\leq \lambda\mathbb{E}\| (\mathbf{I} - \mathbf{P})\Phi_K^{t-1}\| + 2\mathbb{E}\|\sum_{k=0}^{K-1} \eta_k^t \Gamma_k^t\|.$$

The equality adopts $(\mathbf{A} - \mathbf{P})(\mathbf{I} - \mathbf{P}) = \mathbf{A} - \mathbf{P} - \mathbf{AP} + \mathbf{PP} = \mathbf{A} - \mathbf{PA}$. We know the fact that $\Phi_k^t = 0$ where $(t, k) \in (t_0, k_0)$. Thus unwinding the above inequality we have:

$$\mathbb{E}\| (\mathbf{I} - \mathbf{P})\Phi_K^t \| \leq \lambda^{t-t_0+1}\mathbb{E}\| (\mathbf{I} - \mathbf{P})\Phi_K^{t_0-1}\| + 2\sum_{s=t_0}^{t} \lambda^{t-s}\mathbb{E}\|\sum_{k=0}^{K-1} \eta_k^s \Gamma_k^s\|$$

$$= 2 \sum_{s=t_0}^{t} \lambda^{t-s} \mathbb{E} \| \sum_{k=0}^{K-1} \eta_k^s \Gamma_k^s \|.$$

To maintain the term of $\mathbf{A}$, we have:

$$(\mathbf{A} - \mathbf{P}) \Phi_K^t = (\mathbf{A} - \mathbf{P}) \mathbf{A} \Phi_K^{t-1} - (\mathbf{A} - \mathbf{P}) \sum_{k=0}^{K-1} \eta_k^t \Gamma_k^t$$

$$= (\mathbf{A} - \mathbf{P}) (\mathbf{A} - \mathbf{P}) \Phi_K^{t-1} - (\mathbf{A} - \mathbf{P}) \sum_{k=0}^{K-1} \eta_k^t \Gamma_k^t.$$

The second equality adopts $(\mathbf{A} - \mathbf{P}) (\mathbf{A} - \mathbf{P}) = (\mathbf{A} - \mathbf{P}) \mathbf{A} - \mathbf{A}\mathbf{P} + \mathbf{P}\mathbf{P} = (\mathbf{A} - \mathbf{P}) \mathbf{A}$. Therefore we have the following recursive formula:

$$\mathbb{E} \| (\mathbf{A} - \mathbf{P}) \Phi_K^t \| \leq \mathbb{E} \| (\mathbf{A} - \mathbf{P}) (\mathbf{A} - \mathbf{P}) \Phi_K^{t-1} \| + \mathbb{E} \| (\mathbf{A} - \mathbf{P}) \sum_{k=0}^{K-1} \eta_k^t \Gamma_k^t \|$$

$$\leq \lambda \mathbb{E} \| (\mathbf{A} - \mathbf{P}) \Phi_K^{t-1} \| + \lambda \mathbb{E} \| \sum_{k=0}^{K-1} \eta_k^t \Gamma_k^t \|.$$

The same as above, we can unwind this recurrence formulation from $t$ to $t_0$ as:

$$\mathbb{E} \| (\mathbf{A} - \mathbf{P}) \Phi_K^t \| \leq \lambda^{t-t_0+1} \mathbb{E} \| (\mathbf{A} - \mathbf{P}) \Phi_K^{t_0-1} \| + \sum_{s=t_0}^{t} \lambda^{t-s+1} \mathbb{E} \| \sum_{k=0}^{K-1} \eta_k^s \Gamma_k^s \|$$

$$= \sum_{s=t_0}^{t} \lambda^{t-s+1} \mathbb{E} \| \sum_{k=0}^{K-1} \eta_k^s \Gamma_k^s \|.$$

Both two terms required to know the upper bound of the accumulation of the gradient differences. In previous works, they often use the **Assumption of the bounded gradient** to upper bound this term as a constant. However, this assumption may not always hold as we introduced in the main text. Therefore, we provide a new upper bound instead. When $(t, k) < (t_0, k_0)$, the sampled data is always the same between the different datasets, which shows $\Gamma_k^t = 0$. When $t = t_0$, only those updates at $k \geq k_0$ are different. When $t > t_0$, all the local gradients difference during local $K$ iterations are non-zero. Thus we can first explore the upper bound of the stages with full $K$ iterations when $t > t_0$. Let the data sample $z$ be the same random data sample and $z/\widetilde{z}$ be a different sample pair for abbreviation, when $t \geq t_0$, we have:

$$\mathbb{E} \| \eta \Gamma_k^t \| = \mathbb{E} \| \eta \left[ g_{0,k}^t - \widetilde{g}_{0,k}^t, g_{1,k}^t - \widetilde{g}_{1,k}^t, \cdots, g_{m,k}^t - \widetilde{g}_{m,k}^t \right]^\top \| \leq \eta \sum_{i \in [m]} \mathbb{E} \| g_{i,k}^t - \widetilde{g}_{i,k}^t \|$$

$$\leq \eta \sum_{i \neq i^\star} \mathbb{E} \| \nabla f_i(w_{i,k}^t, z) - \nabla f_i(\widetilde{w}_{i,k}^t, z) \| + \frac{(S-1)\eta}{S} \mathbb{E} \| \nabla f_{i^\star}(w_{i^\star,k}^t, z) - \nabla f_{i^\star}(\widetilde{w}_{i^\star,k}^t, z) \|$$

$$+ \frac{\eta}{S} \mathbb{E} \| \nabla f_{i^\star}(w_{i^\star,k}^t, z) - \nabla f_{i^\star}(\widetilde{w}_{i^\star,k}^t, z) + \nabla f_{i^\star}(\widetilde{w}_{i^\star,k}^t, z) - \nabla f_{i^\star}(\widetilde{w}_{i^\star,k}^t, \widetilde{z}) \|$$

$$\leq \eta L \sum_{i \neq i^\star} \mathbb{E} \| w_{i,k}^t - \widetilde{w}_{i,k}^t \| + \frac{(S-1)\eta}{S} \mathbb{E} \| w_{i^\star,k}^t - \widetilde{w}_{i^\star,k}^t \| + \frac{\eta}{S} \mathbb{E} \| w_{i^\star,k}^t - \widetilde{w}_{i^\star,k}^t \|$$

$$+ \frac{\eta}{S} \mathbb{E} \| \left( \nabla f_{i^\star}(\widetilde{w}_{i^\star,k}^t, z) - \nabla f_{i^\star}(\widetilde{w}_{i^\star,k}^t) \right) - \left( \nabla f_{i^\star}(\widetilde{w}_{i^\star,k}^t, \widetilde{z}) - \nabla f_{i^\star}(\widetilde{w}_{i^\star,k}^t) \right) \|$$

$$\leq \eta L \sum_{i \in [m]} \mathbb{E} \| w_{i,k}^t - \widetilde{w}_{i,k}^t \| + \frac{2\eta \sigma_l}{S} = \eta L \left( \Delta_k^t + \frac{2\sigma_l}{SL} \right).$$

According to the Lemma 8, 9 and Eq.(20), we bound the gradient difference as:

$$\mathbb{E} \| \eta \Gamma_k^t \| \leq \eta L \left( \Delta_k^t + \frac{2\sigma_l}{SL} \right) \leq \left( \frac{\tau}{\tau_0} \right)^{\mu L} \frac{2\mu \sigma_l}{\tau S}.$$

where $\tau = tK + k$.

Unwinding the summation on $k$ and adopting Lemma 10, we have:

$$\sum_{s=t_0}^{t} \lambda^{t-s} \mathbb{E}\| \sum_{k=0}^{K-1} \eta_k^s \Gamma_k^s \| \le \sum_{s=t_0}^{t} \lambda^{t-s} \sum_{k=0}^{K-1} \mathbb{E}\|\eta_k^s \Gamma_k^s\| \le \frac{2\mu\sigma_l}{S\tau_0^{\mu L}} \sum_{s=t_0}^{t} \lambda^{t-s} \sum_{k=0}^{K-1} \frac{\tau^{\mu L}}{\tau}$$

$$\le \frac{2\mu\sigma_l}{S\tau_0^{\mu L}} \sum_{s=t_0}^{t} \lambda^{t-s} \sum_{k=0}^{K-1} \frac{(sK)^{\mu L}}{sK} = \frac{2\mu\sigma_l}{S} \left(\frac{K}{\tau_0}\right)^{\mu L} \sum_{s=t_0}^{t} \frac{\lambda^{t-s}}{s^{1-\mu L}}$$

$$\le \frac{2\mu\sigma_l}{S} \left(\frac{K}{\tau_0}\right)^{\mu L} \sum_{s=t_0-1}^{t-1} \frac{\lambda^{t-s-1}}{(s+1)^{1-\mu L}} \le \frac{2\mu\sigma_l\kappa_\lambda}{S} \left(\frac{K}{\tau_0}\right)^{\mu L} \frac{1}{t^{1-\mu L}}.$$

Therefore, we get an upper bound on the aggregation gap which is related to the spectrum gap:

$$\mathbb{E}\| (\mathbf{I} - \mathbf{P}) \Phi_K^t \| \le 2 \sum_{s=t_0}^{t} \lambda^{t-s} \mathbb{E}\| \sum_{k=0}^{K-1} \eta_k^s \Gamma_k^s \| \le \frac{4\mu\sigma_l\kappa_\lambda}{S} \left(\frac{K}{\tau_0}\right)^{\mu L} \frac{1}{t^{1-\mu L}}, \tag{21}$$

$$\mathbb{E}\| (\mathbf{A} - \mathbf{P}) \Phi_K^t \| \le \sum_{s=t_0}^{t} \lambda^{t-s+1} \mathbb{E}\| \sum_{k=0}^{K-1} \eta_k^s \Gamma_k^s \| \le \frac{2\mu\sigma_l\lambda\kappa_\lambda}{S} \left(\frac{K}{\tau_0}\right)^{\mu L} \frac{1}{t^{1-\mu L}}. \tag{22}$$

The first inequality provides the upper bound between the difference between the averaged state and the vanilla state, and the second inequality provides the upper bound between the aggregated state and the averaged state.

### B.3.2 STABILITY BOUND

Now, rethinking the update rules in one round and we have:

$$\sum_{i\in[m]} \mathbb{E}\|w_{i,K}^{t+1} - \widetilde{w}_{i,K}^{t+1}\|$$

$$= \sum_{i\in[m]} \mathbb{E}\| \left(w_{i,0}^{t+1} - \widetilde{w}_{i,0}^{t+1}\right) - \sum_{k=0}^{K-1} \eta_k^t \left(g_{i,k}^t - \widetilde{g}_{i,k}^t\right) \|$$

$$= \sum_{i\in[m]} \mathbb{E}\| \left(w_{i,0}^{t+1} - \widetilde{w}_{i,0}^{t+1}\right) - \left(w_{i,K}^t - \widetilde{w}_{i,K}^t\right) + \left(w_{i,K}^t - \widetilde{w}_{i,K}^t\right) - \sum_{k=0}^{K-1} \eta_k^t \left(g_{i,k}^t - \widetilde{g}_{i,k}^t\right) \|$$

$$\le \sum_{i\in[m]} \left[ \mathbb{E}\| \left(w_{i,0}^{t+1} - \widetilde{w}_{i,0}^{t+1}\right) - \left(w_{i,K}^t - \widetilde{w}_{i,K}^t\right) \| + \mathbb{E}\| \left(w_{i,K}^t - \widetilde{w}_{i,K}^t\right) \| + \mathbb{E}\| \sum_{k=0}^{K-1} \eta_k^t \left(g_{i,k}^t - \widetilde{g}_{i,k}^t\right) \| \right]$$

$$\le \sum_{i\in[m]} \mathbb{E}\| \left(w_{i,K}^t - \widetilde{w}_{i,K}^t\right) \| + m\mathbb{E}\left[ \frac{1}{m} \sum_{i\in[m]} \| \left(w_{i,0}^{t+1} - \widetilde{w}_{i,0}^{t+1}\right) - \left(w_{i,K}^t - \widetilde{w}_{i,K}^t\right) \| \right]$$

$$+ \sum_{i\in[m]} \mathbb{E}\| \sum_{k=0}^{K-1} \eta_k^t \left(g_{i,k}^t - \widetilde{g}_{i,k}^t\right) \|$$

$$\le \sum_{i\in[m]} \mathbb{E}\| \left(w_{i,K}^t - \widetilde{w}_{i,K}^t\right) \| + m\mathbb{E}\sqrt{ \frac{1}{m} \sum_{i\in[m]} \| \left(w_{i,0}^{t+1} - \widetilde{w}_{i,0}^{t+1}\right) - \left(w_{i,K}^t - \widetilde{w}_{i,K}^t\right) \|^2 }$$

$$+ \sum_{i\in[m]} \mathbb{E}\| \sum_{k=0}^{K-1} \eta_k^t \left(g_{i,k}^t - \widetilde{g}_{i,k}^t\right) \|$$

$$= \sum_{i\in[m]} \mathbb{E}\| \left(w_{i,K}^t - \widetilde{w}_{i,K}^t\right) \| + \sqrt{m}\mathbb{E}\|\Phi_0^{t+1} - \Phi_K^t\| + \sum_{i\in[m]} \mathbb{E}\| \sum_{k=0}^{K-1} \eta_k^t \left(g_{i,k}^t - \widetilde{g}_{i,k}^t\right) \|$$

$$
\begin{aligned}
&= \sum_{i \in [m]} \mathbb{E}\| \left( w_{i,K}^t - \widetilde{w}_{i,K}^t \right) \| + \sqrt{m}\mathbb{E}\|\mathbf{A}\Phi_K^t - \Phi_K^t\| + \sum_{i \in [m]} \mathbb{E}\| \sum_{k=0}^{K-1} \eta_k^t \left( g_{i,k}^t - \widetilde{g}_{i,k}^t \right) \| \\
&\leq \sum_{i \in [m]} \mathbb{E}\| \left( w_{i,K}^t - \widetilde{w}_{i,K}^t \right) \| + \sqrt{m}\mathbb{E}\| \left( \mathbf{A} - \mathbf{P} \right) \Phi_K^t\| + \sqrt{m}\mathbb{E}\| \left( \mathbf{P} - \mathbf{I} \right) \Phi_K^t\| \\
&\quad + \sum_{i \in [m]} \mathbb{E}\| \sum_{k=0}^{K-1} \eta_k^t \left( g_{i,k}^t - \widetilde{g}_{i,k}^t \right) \|.
\end{aligned}
$$

Therefore, we can bound this by two terms in one complete communication round. One is the process of local $K$ SGD iterations, and the other is the aggregation step. For the local training process, we can continue to use Lemma 6, 7, and 9. Let $\tau = tK + k$ as above, we have:

$$
\begin{aligned}
&\Delta_K^t + \frac{2\sigma_l}{SL} \\
&\leq \left[ \prod_{k=0}^{K-1} \left( 1 + \eta_k^t L \right) \right] \left( \Delta_0^t + \frac{2\sigma_l}{SL} \right) = \left[ \prod_{k=0}^{K-1} \left( 1 + \frac{\mu L}{\tau} \right) \right] \left( \Delta_0^t + \frac{2\sigma_l}{SL} \right) \\
&\leq \left[ \prod_{k=0}^{K-1} e^{\frac{\mu L}{\tau}} \right] \left( \Delta_0^t + \frac{2\sigma_l}{SL} \right) = e^{\mu L \sum_{k=0}^{K-1} \frac{1}{\tau}} \left( \Delta_0^t + \frac{2\sigma_l}{SL} \right) \\
&\leq e^{\mu L \ln\left( \frac{t+1}{t} \right)} \left( \Delta_0^t + \frac{2\sigma_l}{SL} \right) = \left( \frac{t}{t-1} \right)^{\mu L} \left( \Delta_0^t + \frac{2\sigma_l}{SL} \right) \\
&\leq \left( \frac{t}{t-1} \right)^{\mu L} \left[ \Delta_K^{t-1} + \sqrt{m} \left( \mathbb{E}\| \left( \mathbf{A} - \mathbf{P} \right) \Phi_K^t\| + \mathbb{E}\| \left( \mathbf{P} - \mathbf{I} \right) \Phi_K^t\| \right) + \frac{2\sigma_l}{SL} \right] \\
&\leq \left( \frac{t}{t-1} \right)^{\mu L} \left( \Delta_K^{t-1} + \frac{2\sigma_l}{SL} \right) + \sqrt{m} \left( \frac{t}{t-1} \right)^{\mu L} \left( \mathbb{E}\| \left( \mathbf{A} - \mathbf{P} \right) \Phi_K^t\| + \mathbb{E}\| \left( \mathbf{P} - \mathbf{I} \right) \Phi_K^t\| \right) \\
&\leq \underbrace{\left( \frac{t}{t-1} \right)^{\mu L} \left( \Delta_K^{t-1} + \frac{2\sigma_l}{SL} \right)}_{\text{local updates}} + \underbrace{\frac{6\sqrt{m}\mu\sigma_l\kappa_\lambda}{S} \left( \frac{K}{\tau_0} \right)^{\mu L} \left( \frac{t}{t-1} \right)^{\mu L} \frac{1}{t^{1-\mu L}}}_{\text{aggregation gaps}},
\end{aligned}
$$

The last adopts the Eq.(21) and (22), and the fact $\lambda \leq 1$.

Obviously, in the decentralized federated learning setup, the first term still comes from the updates of the local training. The second term comes from the aggregation gaps, which is related to the spectrum gap $\lambda$. Unwinding this from $t_0$ to $T$, we have:

$$
\begin{aligned}
\Delta_K^T + \frac{2\sigma_l}{SL} &\leq \left( \frac{TK}{\tau_0} \right)^{\mu L} \frac{2\sigma_l}{SL} + \frac{6\sqrt{m}\mu\sigma_l\kappa_\lambda}{S} \left( \frac{K}{\tau_0} \right)^{\mu L} \sum_{t=t_0+1}^{T} \left( \frac{t}{t-1} \right)^{\mu L} \frac{1}{t^{1-\mu L}} \\
&\leq \left( \frac{TK}{\tau_0} \right)^{\mu L} \frac{2\sigma_l}{SL} + \frac{12\sqrt{m}\mu\sigma_l\kappa_\lambda}{S} \left( \frac{K}{\tau_0} \right)^{\mu L} \sum_{t=t_0+1}^{T} \frac{1}{t^{1-\mu L}} \\
&\leq \left( \frac{TK}{\tau_0} \right)^{\mu L} \frac{2\sigma_l}{SL} + \frac{12\sqrt{m}\mu\sigma_l\kappa_\lambda}{S} \left( \frac{K}{\tau_0} \right)^{\mu L} \frac{t^{\mu L}}{\mu L} \bigg|_{t=t_0+1}^{t=T} \\
&\leq \left( \frac{TK}{\tau_0} \right)^{\mu L} \frac{2 \left( 1 + 6\sqrt{m}\kappa_\lambda \right) \sigma_l}{SL}.
\end{aligned}
$$

The second inequality adopts the fact that $1 < \frac{t}{t-1} \leq 2$ when $t > 1$ and the fact of $0 < \mu < \frac{1}{L}$.

According to the Lemma 5, the first term in the stability (conditions is omitted for abbreviation) can be bounded as:

$$
\mathbb{E}\|w^{T+1} - \widetilde{w}^{T+1}\| \leq \frac{1}{m} \sum_{i \in [m]} \mathbb{E}\| \left( w_{i,K}^T - \widetilde{w}_{i,K}^T \right) \| \leq \left( \frac{TK}{\tau_0} \right)^{\mu L} \frac{2 \left( 1 + 6\sqrt{m}\kappa_\lambda \right) \sigma_l}{mSL}.
$$

Therefore, we can upper bound the stability in decentralized federated learning as:

$$\mathbb{E}\left[|f(w^{T+1}; z) - f(\widetilde{w}^{T+1}; z)|\right] \leq G\mathbb{E}\left[\|w^{T+1} - \widetilde{w}^{T+1}\| \mid \xi\right] + \frac{U\tau_0}{S}$$

$$\leq \frac{2\sigma_l G}{SL}\left(\frac{1 + 6\sqrt{m}\kappa_\lambda}{m}\right)\left(\frac{TK}{\tau_0}\right)^{\mu L} + \frac{U\tau_0}{S}.$$

The same as the centralized setup, to minimize the error of the stability, we can select a proper event $\xi$ with a proper $\tau_0$. For $\tau \in [1, TK]$, by selecting $\tau_0 = \left(\frac{2\sigma_l G}{UL}\frac{1+6\sqrt{m}\kappa_\lambda}{m}\right)^{\frac{1}{1+\mu L}}(TK)^{\frac{\mu L}{1+\mu L}}$, we get the minimal:

$$\mathbb{E}\left[|f(w^{T+1}; z) - f(\widetilde{w}^{T+1}; z)|\right] \leq \frac{2U\tau_0}{S} = \frac{2U}{S}\left(\frac{2\sigma_l G}{UL}\frac{1+6\sqrt{m}\kappa_\lambda}{m}\right)^{\frac{1}{1+\mu L}}(TK)^{\frac{\mu L}{1+\mu L}}$$

$$= \frac{4}{S}\left(\frac{\sigma_l G}{L}\right)^{\frac{1}{1+\mu L}}\left(\frac{1+6\sqrt{m}\kappa_\lambda}{m}\right)^{\frac{1}{1+\mu L}}(UTK)^{\frac{\mu L}{1+\mu L}}.$$

