# OpenReview forum: "Which mode is better for federated learning? Centralized or Decentralized"
_ICLR.cc/2024/Conference — Submitted to ICLR 2024_

### Official Review · Reviewer_3qgW · 2023-10-30

**Soundness:** 3 good
**Presentation:** 1 poor
**Contribution:** 3 good
**Rating:** 6
**Confidence:** 3

**Summary:**

The study examines the performances of both Fed-Avg and decentralized local SGD, known as DFedAvg, in terms of excess risk and generalization, which differs from the conventional convergence rate analysis. Results indicate that centralized methods consistently outperform decentralized ones in terms of generalization. Additionally, the research reveals a requisite network topology condition for DFedAvg, and if not met, the generalization becomes worse.

**Strengths:**

The paper provides a theoretical analysis of the performance differences between centralized and decentralized methods, focusing on excess risk and generalization. Its strength is in the conclusion that CFL always generalizes better than DFL. This provides an explanation as to why CFL has better empirical results than DFL in deep learning applications.

**Weaknesses:**

Many related works are missing, and a more thorough literature review is necessary.

Several related studies are not included, necessitating a more comprehensive literature review.

The evaluation focuses solely on the DFedAvg method, which is known to have inherent biases. Various improved versions exist, such as local gradient tracking and local exact-diffusion, which might yield different outcomes. For the Fed-Avg method, one might also contemplate incorporating Scaffold.


The presentation of the results is challenging to comprehend and necessitates improved organization and clarity.

**Questions:**

Table 1 is presented without an explanation of its significance and implications in the introduction, making it necessary for readers to go through the entire paper to understand the table.


This work seems to lack a thorough review of related literature. For example, attributing the DFedAvg framework solely to Sun et al. (2022) is inaccurate. Multiple studies have examined the scenario of local updates in decentralized training, including:

-	Wang, Jianyu, and Gauri Joshi. "Cooperative SGD: A unified framework for the design and analysis of local-update SGD algorithms." The Journal of Machine Learning Research 22.1 (2021): 9709-9758.
-	Koloskova, Anastasia, et al. "A unified theory of decentralized sgd with changing topology and local updates." International Conference on Machine Learning. PMLR, 2020.

Similarly, numerous studies have explored DSGD, its model consistency, and have proposed corrective methods, including:

-	Yuan, Kun, et al. "On the influence of bias-correction on distributed stochastic optimization." IEEE Transactions on Signal Processing 68 (2020): 4352-4367.
-	Shi, Wei, et al. "Extra: An exact first-order algorithm for decentralized consensus optimization." SIAM Journal on Optimization 25.2 (2015): 944-966.
-	Alghunaim, Sulaiman A., and Kun Yuan. "A unified and refined convergence analysis for non-convex decentralized learning." IEEE Transactions on Signal Processing 70 (2022): 3264-3279.
-	Mishchenko, Konstantin, et al. "Proxskip: Yes! local gradient steps provably lead to communication acceleration! finally!." International Conference on Machine Learning. PMLR, 2022.
-	Nguyen, Edward Duc Hien, et al. "On the performance of gradient tracking with local updates." arXiv preprint arXiv:2210.04757 (2022).
-	Alghunaim, Sulaiman A. "Local Exact-Diffusion for Decentralized Optimization and Learning." arXiv preprint arXiv:2302.00620 (2023).


How does the generalization analysis presented in this study differ from that of DSGD without local updates as detailed in:


-	Taheri, Hossein, and Christos Thrampoulidis. "On generalization of decentralized learning with separable data." International Conference on Artificial Intelligence and Statistics. PMLR, 2023.
-	Bars, Batiste Le, Aurélien Bellet, and Marc Tommasi. "Improved Stability and Generalization Analysis of the Decentralized SGD Algorithm." arXiv preprint arXiv:2306.02939 (2023).
-	Zhu, Tongtian, et al. "Topology-aware generalization of decentralized sgd." International Conference on Machine Learning. PMLR, 2022.

Furthermore, the bounds presented in theorems 1 and 2 demand a comprehensive explanation. A clearer breakdown of the influence of each parameter is necessary. Specifically, the intuition behind equation (9) where the bound seems to deteriorate with increasing values of T and K remains elusive (to readers not quite familiar with generalization analysis).

What is small $k$ in theorem 1?

In the scenario of a fully connected network, DFedAvg reduces to Fed-Avg. However, table 1 indicates disparate generalization errors for both under this condition. Could you provide clarification on this discrepancy?

---

> ### Author Response · Authors · 2023-11-12
> **Response to Reviewer 3qgW (1/3)**
>
> Thank you very much for your review and affirmation of our work. We'll answer your questions one by one in the following, including some misunderstandings and some essential academic questions worth exploring. We are also very honored to share some of our understandings with you.
>
> ## 1. About the question of "Table 1 is presented without an explanation of its significance and implications":
> Thank you for pointing out this and we are very sorry for the inconvenience with Table 1. We will add a paragraph in the introduction section to describe them in detail.
>
> Actually, Table 1 summarizes the excess risk of FedAvg and DFedAvg. From the excess risk perspective, the best partial participation ratio in CFL is $\frac{n}{m}=\mathcal{O}(m^{-\frac{\mu L}{1+2\mu L}})$ and the best topology in DFL satisfies $\kappa_\lambda=0$. Under their best selections respectively, the generalization in CFL achieves $\mathcal{O}(m^{-\frac{1+\mu L}{1+2\mu L}})$ and the generalization in DFL achieves $\mathcal{O}(m^{-\frac{1}{1+\mu L}})$.
>
>
> ## 2. About the question of "this work lacks a thorough review of related literature, including [1,2]":
> Thank you for pointing out this and we are sorry for causing you misunderstanding. We do not attribute the DFedAvg framework solely to any single work because it has been studied for a long time. [3] summarizes and provides an optimization convergence on the decentralized heterogeneous dataset in the FL framework which is explicitly targeted at DFL scenarios under the general assumptions widely used in the current FL community.
>
> The mentioned paper [2] has been cited in our paper (page 3 paragraph "Decentralized federated learning"). The mentioned paper [1] mainly focuses on the optimization convergence of local-update-SGD under different paradigms. Our paper mainly focuses on the generalization difference between CFL and DFL. We will add [1] to the related work to further summarize the development of the decentralized approaches.
>
>
> [1] Cooperative SGD: A Unified Framework for the Design and Analysis of Local-update SGD Algorithms (JMLR2021)
>
> [2] A Unified Theory of Decentralized sgd with Changing Topology and Local Updates (ICML2020)
>
> [3] Decentralized Federated Averaging (TPAMI)
>
>
> ## 3. About the question of "Similarly, numerous studies have explored DSGD, its model consistency, and have proposed corrective methods, including [1,2,3,4,5,6]":
> I am very grateful to the reviewer for mentioning several extensions of decentralized approaches.
>
> First of all, I would like to clarify the main target of our paper. We do not focus on proposing a new decentralized method. Our study mainly attempts to answer a question mentioned in the introduction: "Which framework is better for federated learning? Centralized or decentralized?". Several studies have learned the convergence rate in decentralized approaches could achieve a comparable optimization convergence rate as the centralized approaches. However, its performance in empirical studies is always not as good as we expected. This motivated us to explore the generalization performance. Therefore, we focus the related work targeted more on the generalization and stability analysis in decentralized methods. But all mentioned papers [1,2,3,4,5,6] focus more on the convergence analysis in their papers.
>
> In order to dispel the reviewer's concerns, we will add a new paragraph in related work to introduce the development of both the centralized and the decentralized approaches including all missed work mentioned in the rebuttal. Thank you again for pointing out this issue.
>
>
> [1] On the influence of bias-correction on distributed stochastic optimization
>
> [2] Extra: An exact first-order algorithm for decentralized consensus optimization
>
> [3] A unified and refined convergence analysis for non-convex decentralized learning
>
> [4] Proxskip: Yes! local gradient steps provably lead to communication acceleration! finally!
>
> [5] On the performance of gradient tracking with local updates
>
> [6] Local Exact-Diffusion for Decentralized Optimization and Learning

---

> ### Author Response · Authors · 2023-11-12
> **Response to Reviewer 3qgW (2/3)**
>
> ## 4. About the question of "How does the generalization analysis presented in this study differ from that of DSGD without local updates as detailed in [1,2,3]:":
> We summarize the difference as follows.
>
> (1) All analysis in [1] only is suitable for the **convex** function (Assumption 2 in their paper.) In their conclusions, they mentioned "We believe our work opens several directions, with perhaps the most exciting one being the analysis of non-convex objectives.". And, Our analysis targets the smooth and **non-convex** objectives.
>
> (2) To compare to [2], we provide a table in the following.
> |       |   Lipschitz Function Required   | Related to Spectrum |
> |:-----:|:-----:|:-----:|
> | [2] | $\sqrt{}$ | $\times$ |
> | our paper | $\times$ | $\sqrt{}$ |
>
> First of all, in our proofs, we remove the strong assumption of the Lipschitz function (which indicates the gradient must be upper bounded) in the stability analysis. Bounded gradients force each iteration to be still upper bounded. However, many previous studies learn that this assumption may not always hold, especially in current deep learning models. Our proofs indicate that even the iterations are not necessarily bounded, the final stability is still bounded.
>
> Furthermore, in their Theorem 3.1, 3.2, and 3.3, the stability bound **is not related to the spectral gap or information of topology**. One of the main targets of the research in decentralized methods is to understand the impact of topology on the final performance. Our paper provides the stability analysis with the topology coefficient, which is dominated by $\mathcal{O}(\left(\frac{1+6\sqrt{m}\kappa_\lambda}{m}\right)^\frac{1}{1+\mu L})$, where $m$ is the number of the total clients and $\kappa_\lambda$ is the coefficient related to the spectral gap $\lambda$. Furthermore, we derive the dominant factors of the partial participation in FL to be $\mathcal{O}(\frac{n^\frac{\mu L}{1+\mu L}}{m})$. These theorems in our paper can fairly compare the performance differences between CFL and DFL under the same assumptions.
>
> (3) [3] not only must rely on the strong assumption of the Lipschitz function, but also require a additional assumption: "the weight differences at the $t$-th iteration are multivariate normally distributed, $w_{1,k}^t-w_{2,k}^t\sim\mathcal{N}(\mu_k^t,\sigma_{t,k}^tI_d)$" (Assumption A.4 in their paper). This assumption is too strong for the stability analysis because it forces the parameters of all layers to be normalized bounded. Compared with their work, we do not require any extra assumptions on the stability analysis, which could be more general in the current deep learning community.
>
> [1] On generalization of decentralized learning with separable data
>
> [2] Improved Stability and Generalization Analysis of the Decentralized SGD Algorithm
>
> [3] Topology-aware generalization of decentralized sgd
>
>
> ## 5. About the question of "Furthermore, the bounds presented in theorems 1 and 2 demand a comprehensive explanation.":
> Thank you for pointing out this issue and we will add more details to demonstrate our theorem 1 and 2. Actually, the most important part is the active coefficient part (related to $n$, $m$, and $\kappa_\lambda$), which reflects the essential difference between CFL and DFL. Our five main conclusions are also derived from the derivation according to this part. Sorry for the impact on reading, we will add some texts to explain the results of stability analysis, which is proposed in [1].
>
> [1] Train faster, generalize better: Stability of stochastic gradient descent (ICML2016)
>
>
> ## 6. About the question of "What is small $k$ in theorem 1?":
> We will add a notation table in section 3 to introduce all marks in our paper. $k$ is the index of the local training iteration, which is limited within $[1, K]$.

---

> ### Author Response · Authors · 2023-11-12
> **Response to Reviewer 3qgW (3/3)**
>
> ## 7. About the question of "Table 1 indicates disparate generalization errors between under full-connection in DFL and CFL":
> Sorry for the misunderstanding here. In the scenario of a fully connected network, DFedAvg reduces to FedAvg. From our **analysis of generalization**, we can see:
> |   mode    |   CFL (Eq.(9) in our paper)   |  DFL (Eq.(11) in our paper) |
> |:-----:|:-----:|:-----:|
> | general generalization | $\frac{4}{S}\left(\frac{\sigma_lG}{L}\right)^{\frac{1}{1+\mu L}}\left(\frac{n^{\frac{\mu L}{1+\mu L}}}{m}\right)\left(UTK\right)^{\frac{\mu L}{1+\mu L}}$ | $\frac{4}{S}\left(\frac{\sigma_lG}{L}\right)^{\frac{1}{1+\mu L}}\left(\frac{1+6\sqrt{m}\kappa_\lambda}{m}\right)^\frac{1}{1+\mu L}\left(UTK\right)^{\frac{\mu L}{1+\mu L}}$ |
> | when fully connected | $n=m$ |  $\kappa_\lambda=0$ |
> |generalization under fully connected| $\frac{4}{S}\left(\frac{\sigma_lG}{L}\right)^{\frac{1}{1+\mu L}}\left(\frac{1}{m}\right)^\frac{1}{1+\mu L}\left(UTK\right)^{\frac{\mu L}{1+\mu L}}$ | $\frac{4}{S}\left(\frac{\sigma_lG}{L}\right)^{\frac{1}{1+\mu L}}\left(\frac{1}{m}\right)^\frac{1}{1+\mu L}\left(UTK\right)^{\frac{\mu L}{1+\mu L}}$ |
>
> Therefore, the full-connection topology in DFL shows the same generalization performance as all participantion in CFL.
>
> **What is the meaning of Table 1?**
>
> **Table.1 considers the joint performance of excess risk, which is, the sum of optimization errors and generalization errors.** From the perspective of excess risk, we provide the best selection of active ratio ($n$ in CFL) and best topology (in DFL) respectively in Table 1. Under the best selections of each framework, we also show their generalization performance. Table 1 indicates that in FedAvg, from the excess risk perspective, we do not need all clients to participate in per communication round ($n< m$). Only when $n=m$, FedAvg is equal to the DFedAvg under full-connection topology.
>
>
> #### It is a pleasure to discuss this with you, which will help us to further improve this work. We will add all citations in the rebuttal and add a paragraph to summarize the development of decentralized approaches. We compare the mentioned studies with our work in Answer.4. We clarified the misunderstanding of generalization conclusions in Answer.7. If there are any other questions or concerns, we are happy to continue the discussion with you. Thank you again for reading this rebuttal.

---

> > ### Comment · Reviewer_3qgW · 2023-11-19
> >
> > I have read the reviewers response and maintain my original review score, which I believe is fair.

---

> > > ### Author Response · Authors · 2023-11-19
> > > **Re:Reponse to Reviewer 3qgW**
> > >
> > > Thank you for reading our rebuttal and the positive response. Our work provides a fair comparison between the centralized and decentralized training modes and we hope all concerns above have been resolved. Thank you again for the review.

---

### Official Review · Reviewer_4LFR · 2023-11-02

**Soundness:** 1 poor
**Presentation:** 1 poor
**Contribution:** 1 poor
**Rating:** 3
**Confidence:** 4

**Summary:**

This paper investigated the generalization performance of federated sgd under centralized and decentralized settings. However, this paper missed some important references so that the contribution is not clear compared with those existing works.

**Strengths:**

1. Studying the generalization performance of Fedavg and Dfedavg is interesting.

2. The writing is good. It is easy to follow.

**Weaknesses:**

1. There are many existing works studying the generalization of centralized and decentralized federated learning algorithms. However, this paper missed a lot of them. Then, it is not clear what new contributions this paper has. e.g.,is the bound of this paper comparable with existing ones?

[1] https://openreview.net/pdf?id=-EHqoysUYLx

[2] https://arxiv.org/abs/2306.02939

2. This paper does not show how the heterogeneity affects the generalization error.

3. When the communication graph is fully connected, DFedAvg becomes FedAvg. But the generalization error of this paper does not have this relationship.

**Questions:**

1. There are many existing works studying the generalization of centralized and decentralized federated learning algorithms. However, this paper missed a lot of them. Then, it is not clear what new contributions this paper has. e.g.,is the bound of this paper comparable with existing ones?

[1] https://openreview.net/pdf?id=-EHqoysUYLx

[2] https://arxiv.org/abs/2306.02939

2. This paper does not show how the heterogeneity affects the generalization error.

3. When the communication graph is fully connected, DFedAvg becomes FedAvg. But the generalization error of this paper does not have this relationship.

---

> ### Author Response · Authors · 2023-11-12
> **Response to Reviewer 4LFR (1/2)**
>
> Thank you very much for your review and valuable comments on our work. We'll answer your questions one by one in the following, including some misunderstandings and some essential academic questions worth exploring. We are also very honored to share some of our understandings with you.
>
> ## 1. About the question of "it is not clear what new contributions this paper compared with [2,3]":
> (1) **Our contributions**: Firstly thank you for pointing out these related works. There are really many independent studies based on centralized or decentralized approaches. However, to our best knowledge, there is no complete conclusion to indicate which one of them performs better in the FL framework. Since [1] learns D-PSGD could achieve comparable convergence as C-PSGD, it is generally believed that the decentralized methods will achieve the same performance as the centralized methods. However, decentralized experimental performance has always been poor. Therefore, we want to learn their stability under the same general assumptions via the same analysis framework to fairly compare their generalization performance. Our paper completes the further exploration of the comparison within these two modes in FL. Meanwhile, we also find some novel and interesting conclusions based on the stability and excess risk analysis. We summarize them in the following table.
>
> | Main Conclusions from our Proofs |
> | :-----|
> | 1. In CFL, from the excess risk perspective, the best training mode per round is partial participation.  |
> | 2. In CFL, when the local interval increases, we should decrease the active ratios per round. |
> | 3. In DFL, a better topology must satisfy the spectrum coefficient is small enough. |
> | 4. In DFL, to avoid performance collapse, the spectrum coefficient must be lower than $\mathcal{O}(\sqrt{m})$. |
> | 5. From a stability perspective, DFL always generalizes worse than CFL. |
> 1,2: page 6; 3,4: page 7; 5: Table.3
>
>
> (2) Let us summarize a simple comparison between our work and the mentioned work [2,3]. Though they focus on the exploration of centralized or decentralized stability, they are still largely different from our work.
>
> |       |   Lipschitz Function Required   |  Additional Assumptions Required | Comparison between CFL and DFL |
> |:-----:|:-----:|:-----:|:-----:|
> | [2] | $\sqrt{}$ | $\sqrt{}$ |  $\times$ |
> | [3] | $\sqrt{}$ |  $\times$ | $\times$ |
> | our paper | $\times$ | $\times$ | $\sqrt{}$ |
>
> First of all, in our proofs, we remove the strong assumption of the Lipschitz function (which indicates the gradient must be upper bounded) in the stability analysis. Bounded gradients force each iteration to be still upper bounded. However, many previous studies learn that this assumption may not always hold [4,5,6,7], especially in current deep learning models. Our proofs indicate that even the iterations are not necessarily bounded, the final stability is still bounded.
>
> [2] introduces several additional assumptions to analyze the unparticipated clients without bounded losses, but this analysis does not guide us in understanding whether CFL or DFL is better. [3] provides an analysis of D-SGD, but in their Theorem 3.3, the stability bound is not related to the spectral gap or information of topology. One of the main targets of the research in decentralized methods is to understand the impact of topology on the final performance. Our paper provides the stability analysis with the topology coefficient, which is dominated by $\mathcal{O}(\frac{1+6\sqrt{m}\kappa_\lambda}{m})$, where $m$ is the number of the total clients and $\kappa_\lambda$ is the coefficient related to the spectral gap $\lambda$. Furthermore, we derive the dominant factors of the partial participation in FL to be $\mathcal{O}(\frac {n^\frac{\mu L}{1+\mu L}}{m})$. These two theorems in our paper can fairly compare the performance differences between CFL and DFL under the same conditions.
>
>
> [1] Can Decentralized Algorithms Outperform Centralized Algorithms? A Case Study for Decentralized Parallel Stochastic
> Gradient Descent (NIPS2017)
>
> [2] Generalization Bounds for Federated Learning: Fast Rates, Unparticipation Clients and Unbounded Losses (ICLR2023)
>
> [3] Improved Stability and Generalization Analysis of the Decentralized SGD Algorithm (arXiv2023)
>
> [4] The Lipschitz Constant of Self-attention (ICML2021)
>
> [5] Stability and Convergence of Stochastic Gradient Clipping: Beyond Lipschitz Continuity and Smoothness (ICML2021)
>
> [6] Gradient Descent in the Absence of Global Lipschitz Continuity of the Gradients: Convergence, Divergence and Limitations of Its Continuous Approximation. (arXiv)
>
> [7] Beyond Uniform Lipschitz Condition in Differentially Private Optimization (ICML2023)

---

> ### Author Response · Authors · 2023-11-12
> **Response to Reviewer 4LFR (2/2)**
>
> ## 2. About the question of "This paper does not show how the heterogeneity affects the generalization error.":
> Thank you for pointing out this question and it may be a misunderstanding on the stability analysis.
>
> (1) The work [1] mentioned by the reviewer also does not show the heterogeneity impacts. Furthermore, many existing generalization studies in FL also do not involve the heterogeneity impacts [2,3,4].
>
> (2) The stability analysis mainly explores the potential model differences when it is trained on two different datasets. The heterogeneity measures the quality and similarity of the local dataset, which mainly affects the optimization convergence, that is, the efficiency to achieve the optimal state. While stability does not care about the final state. In other words, even if the model converges very poorly on a very poor dataset, it may still maintain good stability because the stability is largely decided by the training algorithms. And the final test accuracy is decided by both the optimization errors and stability errors. Therefore, the quality of the dataset is not important for the stability analysis. That's why most of the previous studies of the generalization analysis in FL do not involve heterogeneity.
>
>
> [1] Generalization Bounds for Federated Learning: Fast Rates, Unparticipation Clients and Unbounded Losses (ICLR2023)
>
> [2] A Robust Federated Learning: The Case of Affine Distribution Shifts (NIPS2020)
>
> [3] Generalized Federated Learning via Sharpness Aware Minimization (ICML2022)
>
> [4] Understanding How Consistency Works in Federated Learning via Stage-wise Relaxed Initialization (NIPS2023)
>
>
> ## 3. About the question of "When the communication graph is fully connected, DFedAvg becomes FedAvg. But the generalization error of this paper does not have this relationship.":
> Thank you for pointing out this and this is a misunderstanding. We will provide a table to introduce their relationships in the following.
>
> |   mode    |   CFL (Eq.(9) in our paper)   |  DFL (Eq.(11) in our paper) |
> |:-----:|:-----:|:-----:|
> | general generalization | $\frac{4}{S}\left(\frac{\sigma_lG}{L}\right)^{\frac{1}{1+\mu L}}\left(\frac{n^{\frac{\mu L}{1+\mu L}}}{m}\right)\left(UTK\right)^{\frac{\mu L}{1+\mu L}}$ | $\frac{4}{S}\left(\frac{\sigma_lG}{L}\right)^{\frac{1}{1+\mu L}}\left(\frac{1+6\sqrt{m}\kappa_\lambda}{m}\right)^\frac{1}{1+\mu L}\left(UTK\right)^{\frac{\mu L}{1+\mu L}}$ |
> | when fully connected | $n=m$ |  $\kappa_\lambda=0$ |
> |generalization under fully connected| $\frac{4}{S}\left(\frac{\sigma_lG}{L}\right)^{\frac{1}{1+\mu L}}\left(\frac{1}{m}\right)^\frac{1}{1+\mu L}\left(UTK\right)^{\frac{\mu L}{1+\mu L}}$ | $\frac{4}{S}\left(\frac{\sigma_lG}{L}\right)^{\frac{1}{1+\mu L}}\left(\frac{1}{m}\right)^\frac{1}{1+\mu L}\left(UTK\right)^{\frac{\mu L}{1+\mu L}}$ |
>
> When the topology is fully connected in DFL, all clients will aggregate the model after the local training. Therefore, it is equivalent to all participation in CFL which indicates $n=m$. Their generalization is the same. Hope this table can avoid this misunderstanding.
>
>
> #### It is a pleasure to discuss this with you, which will help us to further improve this work. We compare in detail the mentioned papers with our paper in Answer.1, explain the issues of heterogeneity terms in Answer.2, and provide a table to avoid misunderstanding in Answer.3. If there are any other questions or concerns, we are happy to continue the discussion with you. Thank you again for reading this rebuttal.

---

> ### Author Response · Authors · 2023-11-22
> **Rebuttal due**
>
> Thank you for the review. Due to rebuttal time constraints, we have not received any response yet. We want to know if our rebuttal resolves the original misunderstanding.

---

### Official Review · Reviewer_9dRK · 2023-11-06

**Soundness:** 2 fair
**Presentation:** 3 good
**Contribution:** 2 fair
**Rating:** 3
**Confidence:** 4

**Summary:**

This work studied the generalization performances of CFL and DFL in terms of excess risks, under the framework of algorithmic stability. They demonstrate theoretically that CFL has superior generalization capabilities and empirically support this claim with experiments.

**Strengths:**

1. Thorough study on CFL and DFL
2. Extensive discussions and experiments.

**Weaknesses:**

1. Authors claimed the study focuses on nonconvex functions (e.g., Table 1), but to determine the final excess risk (Corollary 1.2 and 2.2), the additional PL-condition is imposed, so all in all the study and follow-up discussion comparing CFL and DFL in fact focuses on PL functions only, which restricts the scope of the research. Even though I may agree PL is a bit common in nonconvex literature, but all in all it is still pretty strong (similar to strong convexity though), not to mention the mismatch with deep learning models you used in the experiment part. I think the authors should clearly clarify this point in the theory part, and present it at the beginning of the paper.
2. Follow-up on Corollary 1.2 and 2.2, to derive the final excess risk, the work borrows the work on the optimization literature (Haddadpour & Mahdavi (2019); Zhou et al. (2021)). But as far as I can see, there is a mismatch in the parameter selection between your generalization learning rate and their optimization learning rates (not to mention some assumption differences). Or at least I think authors should have a double check on their text and discussions for a verification. In that sense, I don't think it is correct to simply aggregate the two results together. Could you please clarify this point?
3. In terms of the theoretical proof, as far as I can see, the proof heavily relies on those in Hardt et al., 2016, with some modification in the later recursion to fit the finite-sum structure. Even though the story of the proof is interesting, the theoretical novelty is restricted a bit regarding the resemblance.

All in all, I appreciate the authors' efforts in the work, but I still think the significance of the contribution in the work is still unclear to pinpoint, which requires further clarification. Please definitely indicate if I misunderstood any points in the work. Thank you.

**Questions:**

1. Missing definition of $U$ in Theorem 1, which is the upper bound of function values.
2. Several typos with $||f()-f()||$, which should be revised to absolute values for scalars.

---

> ### Author Response · Authors · 2023-11-12
> **Response to Reviewer 9dRK (1/2)**
>
> Thank you very much for your review and valuable comments on our work. We'll answer your questions one by one in the following, including some misunderstandings and some essential academic questions worth exploring. We are also very honored to share some of our understandings with you.
>
> ## 1. About the question of "All discussions must rely on PL-condition:"
> (1) We first would like to clarify the misunderstanding about the PL-condition and our paper. Our main theoretical contribution is to fairly compare the generalization performance between CFL and DFL via the same analysis framework under the same assumptions. Though we mentioned comparing their excess risk, however, most of our conclusions provided in our paper are based on the stability comparison, **which never adopts the PL-condition.** So we think the comment "so all in all the study and follow-up discussion comparing CFL and DFL in fact focuses on PL functions only" is very inaccurate. Here we make a table in the following to summarize our main novel conclusions and the conditions required.
>
> | Main Conclusions from our Proofs | Use PL-condition or Not |
> | :-----| :----: |
> | 1. In CFL, from the excess risk perspective, the best training mode per round is partial participation.  | $\sqrt{}$ |
> | 2. In CFL, when the local interval increases, we should decrease the active ratios per round. | $\times$ |
> | 3. In DFL, a better topology must satisfy the spectrum coefficient is small enough. | $\times$ |
> | 4. In DFL, to avoid performance collapse, the spectrum coefficient must be lower than $\mathcal{O}(\sqrt{m})$. | $\times$ |
> | 5. From a stability perspective, DFL always generalizes worse than CFL. | $\times$ |
> 1,2: page 6; 3,4: page 7; 5: Table.3
>
> From the above table, we can clearly see that in the five main conclusions in our paper, four of them do not rely on the PL-condition. Therefore, we object to the statement that our discussion is limited to the PL-condition only.
>
>
> (2) In the excess risk, the test error is bounded by the sum of optimization error and generalization error, which is $\varepsilon_e=\varepsilon_{gen}+\varepsilon_{opt}=\mathbb{E}[F(w^T)-f(w^T)] + \mathbb{E}[f(w^T)-f(w^\star)]$. Current non-convex analysis only provides the $\frac{1}{T}\sum_t\Vert\nabla f(w^t)\Vert^2$ term is upper bounded. Therefore, in many studies of analyzing the excess risk, to approximate the bound of $\varepsilon_{opt}=\mathbb{E}[f(w^T)-f(w^\star)]$ term, many additional assumptions are adopted. We follow the studies [1,2] to select the PL-condition to provide the $\varepsilon_{opt}$ term.
>
> [1] Towards Understanding Why Lookahead Generalizes Better Than SGD and Beyond (NIPS2021)
>
> [2] Understanding How Consistency Works in Federated Learning via Stage-wise Relaxed Initialization (NIPS2023)
>
>
>
> ## 2. About the question of "learning rate selections are different in optimization and generalization":
> (1) The same as the answer above, in the five main conclusions in our paper, only one relies on the combination of optimization and generalization errors. Therefore, the other four conclusions about CFL and DFL in the main body of the paper do not face the issue of different selections of the learning rate.
>
> (2) Previous work [1] has succeeded in analyzing the efficiency of the "lookahead" via the excess risk. In Section 4.3 in [1], they introduce the learning rate is selected as $\mathcal{O}(\frac{1}{t})$ in optimization under smoothness and PL-condition in non-convex objectives. In Section D.5 (proof of generalization on page.13) in [1], they clearly introduce that the learning rate is selected as $\mathcal{O}(\frac{1}{t-1})$ in generalization. **The selections of the learning rate are the same in both optimization and generalization.** The same, in our paper, we follow the same selection and let the learning rate be $\mathcal{O}(\frac{1}{t})$ in generalization analysis. In the non-convex federated optimization analysis, under the smoothness and PL-condition, the learning rate is selected as $\mathcal{O}(\frac{\log{t}}{t})\approx\mathcal{O}(\frac{1}{t})$, which is basically consistent with the generalized analysis [2].
>
> [1] Towards Understanding Why Lookahead Generalizes Better Than SGD and Beyond (NIPS2021)
>
> [2] Decentralized Federated Averaging (TPAMI)

---

> ### Author Response · Authors · 2023-11-12
> **Response to Reviewer 9dRK (2/2)**
>
> ## 3. About the question of "the proof heavily relies on those in [2] with some modification in the later recursion":
> We disagree with this point. Uniform stability analysis in deep learning is proposed by [1,2] and has been widely used in a lot of previous studies to analyze various algorithms [3,4,5,6]. As a good analysis framework, many classic conclusions in generalization are obtained based on this analysis. Furthermore, we must emphasize our additional contributions in this work:
>
> (1) To our best knowledge, this is the first work to prove uniform stability without adopting the assumption of Lipschitz continuous function (bounded gradient assumption). Many previous studies have learned the Lipschitz function assumption is too strong in deep models which may not always hold. Our work further refines the uniform stability analysis.
>
> (2) We focus on fairly understanding the difference in generalization performance between CFL and DFL with the help of a uniform stability framework. This is not a simple incremental work. We not only prove that the generalization performance of DFL is always worse than CFL but also find a novel conclusion, that is, in the DFL scenarios, the topology still has a theoretical minimum requirement to avoid the collapse (summarized in the Table in Answer.1). We also conduct empirical studies to validate the conclusions proposed in our paper. Both theoretical analysis and experiments confirm our conclusions.
>
> (3) Beyond [2], we analyze how the topology affects the generalization in DFL, and how active ratio affects the generalization in CFL. In our analysis, we can effectively compare the generalization efficiency of CFL and DFL and give strict theoretical differences in the stability and generalization performance.
>
>
> [1] Stability of Randomized Learning Algorithms. (JMLR 2005)
>
> [2] Train Faster, Generalize Better: Stability of Stochastic Gradient Descent (ICML2016)
>
> [3] Uniform Stability for First-Order Empirical Risk Minimization (NIPS2018)
>
> [4] Fine-Grained Analysis of Stability and Generalization for Stochastic Gradient Descent (ICML 2020)
>
> [5] Towards Understanding Why Lookahead Generalizes Better Than SGD and Beyond (NIPS2021)
>
> [6] Understanding How Consistency Works in Federated Learning via Stage-wise Relaxed Initialization (NIPS2023)
>
>
> ## 4. About the question of "some typos":
> We will fix all $\Vert f() - f()\Vert$ to $\vert f() - f()\vert$ and add a notations table in Section 3 to summarize all marks in this paper. Thank you again for pointing out this issue.
>
>
> #### It is a pleasure to discuss this with you, which will help us to further improve this work. We explain some misunderstanding about the assumption of PL-condition and selection of learning rate in Answer.1 and .2 and summarize the novelty of our work in Answer.3. If there are any other questions or concerns, we are happy to continue the discussion with you. Thank you again for reading this rebuttal.

---

> ### Author Response · Authors · 2023-11-22
> **Rebuttal due**
>
> Thank you for the review. Due to rebuttal time constraints, we have not received any response yet. We want to know if our rebuttal resolves the original misunderstanding.

---

> > ### Comment · Reviewer_9dRK · 2023-11-23
> > **Thank you**
> >
> > Thank you very much for the detailed response.
> >
> > 1. PL-condition
> >
> >    I want to make it clear that I am not discouraging authors' contributions from any perspective, here my concern is such a mismatch makes your writing very confusing to me. Authors never mentioned PL until the very late Corollary 1.2 (and 2.2) in Page 6, while Table 1 (as the highlighted selling point, even if you claimed multiple contributions) comes with the title "**Main results**...... on the **smooth non-convex** objective...", but in fact it is just PL, so I think it is a bit overclaiming. I think such tricky writing should be clarified.
> >
> > 2. Stepsize
> >
> >    I requested authors to "have a double check on their text and discussions for a verification", I am glad that authors provide some further details that authors never mentioned or discussed in the submission (I don't think authors should assume readers know such details).
> >
> >    But all in all, they are different unfortunately, even though authors argued the consistency or they are in the same order. I can imagine that the final result should be as expected. But I think authors should add such detailed discussion into the appendix to make your work self-contained, rather than asking readers to search online.
> >
> >    I think an ideal and reasonable flow should be like, when discussing the excess risk, authors present the "raw" optimization results (without stepsize setting) in the appendix for readers' sanity check, then combine it with your "raw" generalization result, then specify the stepsize to give the final excess risk result in the main text.
> >
> > 3. "To our best knowledge, this is the **first work to prove uniform stability without adopting the assumption of Lipschitz continuous function** (bounded gradient assumption)."
> >
> >    But as far as I understand, your bounded variance in Assumption 2 basically fixed the issue of removing bounded gradients, which Hardt's paper does not take (while implied by bounded gradient), I may view such changes as interesting while a little incremental and as expected.
> >
> >    Also in fact there have already appeared some other works in "uniform stability" with different or milder conditions, for example: Bernstein condition in *Klochkov, Yegor, and Nikita Zhivotovskiy. "Stability and Deviation Optimal Risk Bounds with Convergence Rate $ O(1/n) $." Advances in Neural Information Processing Systems 34 (2021): 5065-5076* (also a work I suggest you can cite). So I insist on my stand.
> >
> > All in all, I acknowledge authors' all five claimed contributions are interesting, but regarding the existing issues, I am still concerned that it may not pass the bar for this venue, and I will keep my score. I still suggest a revision of the writing to better reflect the scope and contribution of the work, and to avoid any overclaim or confusion in the context. Thank you for the effort.

---

> > > ### Author Response · Authors · 2023-11-23
> > > **Thank you for the Response**
> > >
> > > Thank you for the response and we are happy to notice many misunderstandings have been clarified. After reading the responses we have a few extra points to share.
> > >
> > > 1. About the PL-condition. At the very beginning of this paper, we have strengthened that our studies are focusing on the stability and generalization comparison instead of the optimization. We have clearly stated the results of our theoretical analysis in the summary that most of the analysis is irrelated to the PL-condition. We never adopt any "tricky wrinting" to exaggerate our contributions. Sorry to be misleading, but we have emphasized in many places by "from the generalization perspective" to indicate the conditions relied on.
> > >
> > > 2. We will add additional instructions on learning rate selection in the appendix, which is a technique that has been used before [1].  Based on the responses, we realize that the issue of learning rate selection has been clarified.
> > >
> > > [1] Towards Understanding Why Lookahead Generalizes Better Than SGD and Beyond (NIPS2021)
> > >
> > > 3. We are grateful to the reviewer for providing some additional ``different or milder conditions". However, we still want to point out that our study classifies this difference as an essential difference between centralized and decentralized training mode. **It should be noted that it cannot and shouldn't be compensated by other additional assumptions, but is caused by the difference between centralized average and decentralized average.** We are not searching for better assumptions for CFL and DFL and studying whether the performance of each can be improved. **Our core goal is to fairly compare the performance of these two training frameworks.** We believe that this is essential and fundamental research.
> > >
> > > We thank the reviewer for sharing some of the writing tricks and providing some additional assumptions, and the valuable time paid for our work.

---

### Official Review · Reviewer_Fo8N · 2023-11-08

**Soundness:** 3 good
**Presentation:** 3 good
**Contribution:** 3 good
**Rating:** 6
**Confidence:** 3

**Summary:**

The authors proposed the analysis of the uniform stability and excess risk between CFL and DFL. They proved that CFL would give a better generalization performance than DFL. Moreover, they demonstrated that for DFL to serve as an effective compromise, balancing between performance and communication efficiency, its network topology must meet certain minimal criteria. Lastly, the authors provided extensive simulation results to support their theory analysis.

**Strengths:**

1. The paper is well-written and clearly presented.

2. The authors provide a clear comparison of their findings with existing results such as those of vanilla SGD.

3. The authors provide guidelines on how to tune the optimal active numbers and the optimal topology ratio.

**Weaknesses:**

Some notations are not defined in the main body. For example, The matrix U in Theorem 1 is shown before it is formally defined in the Appendix. It would be kind of confusing to readers.

**Questions:**

The paper seems to require the sample gradient to be Lipschitz, according to Lemma 6. However, Assumption 1 requires that the full gradient is Lipschitz. Is this a typo?

---

> ### Author Response · Authors · 2023-11-12
> **Response to Reviewer Fo8N**
>
> Thank you very much for your review and affirmation of our work. We'll answer your questions one by one in the following, including some misunderstandings and some essential academic questions worth exploring. We are also very honored to share some of our understandings with you.
>
> ## 1. About the question of "Some notations are not defined in the main body":
> Thank you for pointing this and we are very sorry that some missing notation definitions have caused difficulty in reading and we will add a comprehensive notation table to clarify each symbol.
>
> ## 2. About the question of "The paper seems to require the sample gradient to be Lipschitz":
> We are very grateful to the reviewer for checking our proof section and we will fix Assumption 1 to a sample gradient formation. Lemma 6 in our paper refers to Lemma 1.1 in [1] and the expansion of Lemma 3.6 in [2]. In the study of stability analysis, Lemma 6 in our paper is a very general recursive formula and many classical works adopt this property to upper bound the difference on the same data sample.
>
>
> [1] Towards Understanding Why Lookahead Generalizes Better Than SGD and Beyond (NIPS2021)
>
> [2] Train faster, generalize better: Stability of stochastic gradient descent (ICML2016)
>
> #### It is a pleasure to discuss this with you, which will help us to further improve this work. We explain some concerns mentioned in the reviews. If there are any other questions or concerns, we are happy to continue the discussion with you. Thank you again for reading this rebuttal.

---

> > ### Comment · Reviewer_Fo8N · 2023-11-22
> > **Thanks for your responses**
> >
> > Thank you for your responses. I chose to maintain my scores.

---

> > > ### Author Response · Authors · 2023-11-23
> > > **Thanks for the review**
> > >
> > > Thank you again for checking the proofs and your valuable time paying for the review. We will update all concerns mentioned to the final version.

---

### Author Response · Authors · 2023-11-17
**Summary of Revisions**

**Thanks to the reviewers for their comments and time paying for our work.**

We have not yet received any feedback from reviewers. We note that some comments may misunderstand some contributions of our work. We therefore would like to correct these points as following summarization.

## Contributions:

1. **We provide a fair stability and generalization comparison between CFL and DFL** under the same general assumptions to understand which one is better. We prove that from the stability perspective, the DFL would always generalize worse than the CFL, which is caused by their essential difference (the topology property).

2. **We remove the assumption of the Lipschitz function required in the vanilla stability analysis**, which further refines the applicability of this conclusion to deep learning models.

3. **We also find and prove two novel properties.** a) In CFL, to achieve better test accuracy, partial participation is better than full participation; b) In DFL, to avoid the performance collapse with the increase of total clients $m$, the spectral coefficient $\kappa_\lambda$ of the topology must be smaller than $\mathcal{O}(\sqrt{m})$.

4. **We draw the conclusion from the theoretical analysis that DFL will always generalize worse than the CFL.** Its strength is to save communication costs across large-scale training. While as a compromise of CFL, it is a trade-off in FL. Our theory also clearly points out the specific magnitude of their generalization differences.

## Some misunderstandings:
1. **``Our analysis must rely on PL-condition.''** This is incorrect. All generalization analysis is irrelated to the PL-condition and our main conclusions all come from the generalization analysis.

2. **``Generalization of full-topology in DFL and full-participation in CFL is different.''** They are the same. We have provided the table to compare and calculate their generalization bound in the separate replies.

**The other misunderstanding we have addressed in detail in the separate replies. We believe these misunderstandings have been well dispelled. If there is still any confusion, we would be honored to continue the discussion.**

Due to limited rebuttal time, we revise our paper based on the first round of review comments, and we summarize the modifications as follows (marked in blue in the paper).

1. We add some sentences to describe the Table 1.

2. We add a paragraph to introduce the development of some decentralized approaches and local-update-based methods in related work.

3. We add a notation table in section 3.

4. We add the definition of the symbol $U$ in two theorems.

5. We correct the typo $\Vert f() - f()\Vert$ to $\vert f() - f() \vert$.

6. We fix the description of the Lipschitz assumption.

7. (**Major**) We expand Table 4 to further and clearly introduce our comparison results between CFL and DFL, including the meaning of some important selections and their corresponding generalization bounds. We believe that Table 4 can effectively eliminate many misunderstandings about this work.

### Thank you again for reading the rebuttal and valuable time paying for our work.

---

### Meta-Review · Area_Chair_F5mJ · 2023-12-10

**Metareview:**

The reviewers had an overall negative sentiment about the paper. The main lukewarm response stems from that there is not a single articulated and well-contextualized result that demonstrates clear novelty in the presence of the extensive FL literature. The presentation can benefit from significant improvement, and a comprehensive, contextualized discussion of how the results in this paper improve the state-of-the-art is currently lacking. Methodology-wise, the technical innovation is limited given Hardt 2016, as the main diff is lifting certain smoothness assumption. As such, the paper is not publishable as it currently stands. I encourage the authors to revise the paper to keep the contribution sharp and highlight the technical innovation and resubmit to the next ML venue.

**Justification For Why Not Higher Score:**

N/A

**Justification For Why Not Lower Score:**

N/A

---

### Decision · Program_Chairs · 2024-01-16

Reject